# A clinically translatable hyperspectral endoscopy (HySE) system for imaging the gastrointestinal tract

Jonghee Yoon [1,2], James Joseph [1,2], Dale J. Waterhouse [1,2], A. Siri Luthman[1,2], George S.D. Gordon[3], Massimiliano di Pietro[4], Wladyslaw Januszewicz[4], Rebecca C. Fitzgerald [4] & Sarah E. Bohndiek [1,2]

Hyperspectral imaging (HSI) enables visualisation of morphological and biochemical information, which could improve disease diagnostic accuracy. Unfortunately, the wide range of image distortions that arise during flexible endoscopy in the clinic have made integration of HSI challenging. To address this challenge, we demonstrate a hyperspectral endoscope (HySE) that simultaneously records intrinsically co-registered hyperspectral and standard-of-care white light images, which allows image distortions to be compensated computationally and an accurate hyperspectral data cube to be reconstructed as the endoscope moves in the lumen. Evaluation of HySE performance shows excellent spatial, spectral and temporal resolution and high colour fidelity. Application of HySE enables: quantification of blood oxygenation levels in tissue mimicking phantoms; differentiation of spectral profiles from normal and pathological ex vivo human tissues; and recording of hyperspectral data under freehand motion within an intact ex vivo pig oesophagus model. HySE therefore shows potential for enabling HSI in clinical endoscopy.

[1] Department of Physics, University of Cambridge, JJ Thomson Avenue, Cambridge CB3 0HE, UK. [2] Cancer Research UK Cambridge Institute, University of Cambridge, Li Ka Shing Centre, Cambridge CB2 0RE, UK. [3] Department of Engineering, University of Cambridge, JJ Thomson Avenue, Cambridge CB3 0FA, UK. [4] MRC Cancer Unit, Hutchison/MRC Research Centre, University of Cambridge, Cambridge CB2 0XZ, UK. Correspondence and requests for materials should be addressed to S.E.B. (email: seb53@cam.ac.uk)

Hyperspectral imaging (HSI), originally developed for remote sensing[1], is an exciting new technique in biomedicine that enables both spatial $(x, y)$ and spectral $(\lambda)$ information to be recorded from a given sample. When applied to tissue diagnostics, the resulting three-dimensional (3D) hypercube of data can encode the properties of light-tissue interactions such as absorption, scattering and fluorescence, which provides label-free biochemical insight into the tissue structure and function[2,3]. HSI has already shown potential in biomedicine: to resolve multiplexed fluorescence signals[4]; measure tissue oxygen levels[5–9]; diagnose both retinal diseases[10–13] and cancers[14–18]; and to visualise tissue function in real-time during surgery[19,20].

Endoscopic imaging is widely used in screening and surveillance of the aerodigestive tract, particularly for the identification of early signs of cancer. The standard-of-care is white light endoscopy (WLE), in which the lumen of an organ is illuminated with white light and the image is captured using a standard colour camera, sensitive only to red, green and blue spectral components, replicating the vision of the human eye. Unfortunately, the subtle changes that arise in pre-cancerous and early cancerous lesions of the gastrointestinal (GI) tract (e.g., colonic adenomas or Barrett's oesophagus dysplasia) are difficult to identify, even with high definition WLE or narrow band imaging, leading to a reported sensitivity of up to only 64% and a high miss rate[21–26]. Given the diagnostic potential shown by point-based spectroscopy techniques in differentiating inflammatory conditions, early dysplastic changes and cancer in the GI tract[27–29], the application of HSI to endoscopic imaging has the potential to substantially improve sensitivity for early cancer diagnostics.

Performing HSI in flexible GI endoscopy has, however, been limited by the complexities of the instrumentation for HSI data acquisition, as well as the practical data reconstruction challenges associated with the wide range of geometric distortions encountered during normal endoscopy operation. There are three key methods through which a data hypercube can normally be obtained: spatial scanning; spectral scanning; or snap-shot imaging[2]. Spatial scanning methods illuminate a single point or line on the sample and exploit dispersion materials such as prisms or gratings to disperse light according to wavelength onto a one-dimensional (1D) or two-dimensional (2D) detector, respectively; the point or line is then scanned to obtain the hypercube. Spatial scanning provides data at high spectral resolution, but traditionally has several limitations: spatial resolution is traded against temporal resolution; it typically requires defined (rigid) mechanical scanning of sample or imaging system[7,16] to acquire a full data set; and offline image processing is needed to reconstruct the hypercube. Spectral scanning methods use tunable bandpass filters, such as liquid crystal or acoustic-optic tunable filters, placed after the light source or at the detector[30]. Spectral scanning avoids mechanical movements of the sample or imaging system, but now trades spectral resolution against temporal resolution and often suffers from low optical throughput due to the use of polarising optics[2,31]. The images recovered from conventional spatial and spectral scanning methods are typically challenging to overlay onto standard white light images as geometric distortions present during data acquisition degrade HSI data[32]. Snap-shot methods use multicolour filters[33] and dispersive techniques[10,15,34] to obtain HSI data simultaneously without scanning, avoiding the challenges of white light image co-registration; these methods are therefore fast, compact, and robust but are usually severely limited in both spatial and spectral resolution[2].

Several HSI endoscopes have been reported for applications in the lungs[17], stomach[35], larynx[36,37], liver[38] and brain[39]. These have generally exploited a tunable filter to convert a conventional WLE to HSI endoscopy, requiring long exposure times for sufficient signal-to-noise level, leading to slow data acquisition

speeds (e.g., over 20 s for 80 spectral channels) and limiting potential for clinical translation, where real-time acquisition is needed. HSI endoscopes with spatial scanning approaches have been developed using a rotational motor and rigid fibre[40] or specialised imaging fibre bundle[41], which have shown proof-of-concept by measuring test targets. Video-rate multispectral imaging endoscopy has been achieved using image mapping spectroscopy[15], but suffers very limited spectral resolution and channel numbers. For routine clinical application of an HSI endoscopy system, real-time data acquisition speed together with broad, dynamic wavelength measurement at high spatial and spectral resolution are required. To maximise sensitivity for early cancer, deployment of HSI in endoscopy would be possible in a freehand imaging environment in a lumen, a demand that is not satisfied using standard implementations of the aforementioned methods.

Here, we describe a method for hyperspectral data acquisition in endoscopy that exploits the benefits of line-scanning HSI and overcomes the limitations by performing co-registered wide-field white light imaging to accurately reconstruct an HSI data cube (hypercube) from freehand operation, without requirement for rigid mechanical scanning. The measurement of both hyperspectral and white light data simultaneously with the resulting hyperspectral endoscope (HySE) enables the use of white light images to correct for unknown geometric distortions encountered during freehand endoscopy, which would otherwise prevent accurate reconstruction of the hypercube. We demonstrate the proof-of-concept for our approach using technical validation in simulations and experiments, combined with biological validation in simple tissue structures, phantoms, ex vivo human tissues, and an intact ex vivo pig oesophagus model. Our flexible HySE is built around a Conformité Européene (CE)-marked babyscope platform[42], which can be inserted into the accessory channel of a standard commercial endoscope, in readiness for future clinical application as a versatile tool for label-free disease identification in the GI tract and lungs.

## Results

**HySE design**. To enable HSI to be performed in a flexible endoscope, we designed an optical system that records a wide-field WLE 2D image $(x, y)$ with an intrinsically co-registered 2D line-scan HSI $(y, \lambda)$ simultaneously (Fig. 1). The natural freehand motion of the endoscope is then exploited to acquire the additional spatial dimension (x) for reconstruction of a full 3D hypercube.

The HySE system is built around a commercial CE-marked babyscope platform that can be inserted through the accessory channel of a standard endoscope to facilitate clinical translation. Illumination is delivered either internally (a white light-emitting diode (LED)), through the endoscope illumination fibre, or externally (a halogen white light source) to the sample. The external illumination was exploited for the experiments of system characterisation and ex vivo tissue measurements to enable a wide-area illumination, whereas the internal illumination was used in freehand imaging and ex vivo pig oesophagus imaging to demonstrate practical applicability in clinical conditions (see Methods). The optical image of the sample, which is placed at the distal end of the endoscope, is relayed to the image acquisition system positioned at the proximal end of the endoscope through a 10,000 fibrelet imaging fibre bundle. The proximal end of the fibre bundle is imaged and magnified by an infinity corrected objective lens before being divided at a beam splitter: 10% of the optical power is delivered to a monochrome or colour camera (wCam) for wide-field white light imaging; the remaining 90% is delivered to the imaging spectrograph, which records line-scan HSI data.

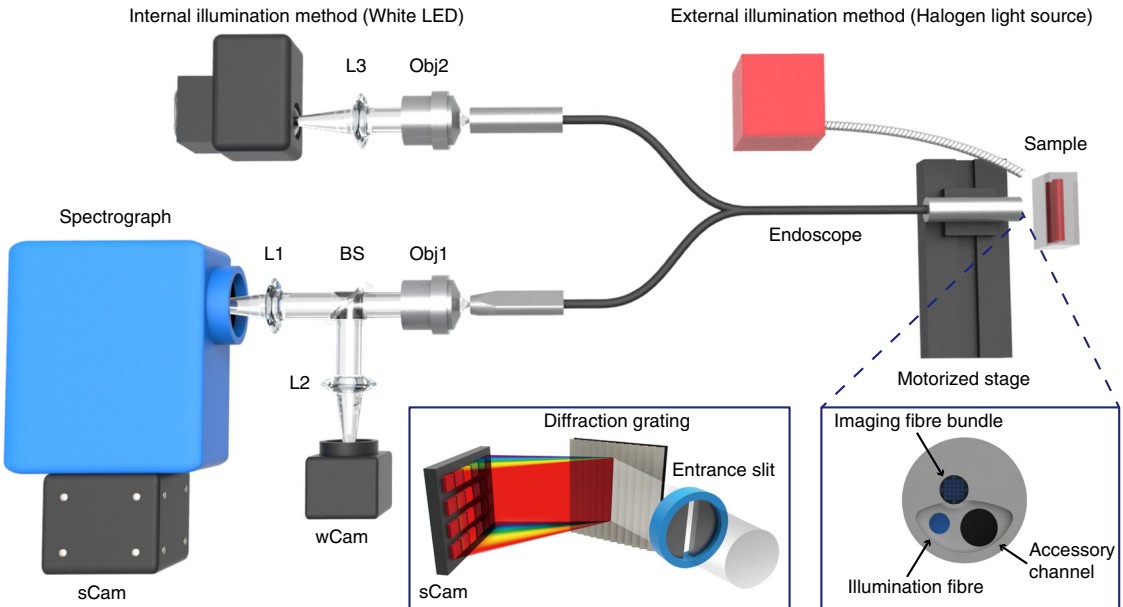

**Fig. 1** Optical design of the hyperspectral endoscope. The system is assembled using a clinically approved endoscope with an integrated illumination fibre, to deliver optical excitation to the sample at the distal end of the endoscope, and an imaging fibre bundle, to relay image information back to the proximal end of the endoscope. Illumination is provided either by coupling a white LED to the illumination fibre (internal illumination method), or by directly illuminating a halogen light source to the scene (external illumination method). Line-scan hyperspectral data are acquired using sCam, a CCD coupled to the spectrograph, whereas wide-field image data are acquired using wCam, a monochrome or colour CMOS camera for image registration purposes. Left inset shows the arrangement of the spectrometer and sCam. Right inset shows details of the distal end of the endoscope. CCD charge coupled device, CMOS complementary metal oxide semiconductor, L1–3 lens, Obj1–2 objective lens, BS beam splitter

The spectrograph entrance slit defines the dimensions of a narrow rectangular area (slit width can be adjusted manually from 10 μm to 3 mm) along the length of the 2D image of the fibre bundle, retaining 1D spatial information ($y$). The reflection grating inside the spectrograph then disperses the light at every point in the 1D spatial ($y$) axis into respective wavelength components (Fig. 1, left inset). The 2D spectrograph camera (sCam) hence records one spatial dimension ($y$) and one spectral dimension ($\lambda$). Three gratings (50, 150 and 300 lines/mm) were tested in this study. The 300 lines/mm grating was used for most of system characterisation experiments and ex vivo tissue measurements to maximise the spectral resolution of the results, whereas the 50 lines/mm grating was exploited for freehand imaging and ex vivo pig oesophagus measurements to obtain a wide spectral bandwidth with high speed operation. As the distal end of the endoscope is moved across the sample, recording the second spatial dimension ($x$), simultaneous line-scan HSI and wide-field imaging enables accurate recovery of the hyperspectral data cube ($x$, $y$, $\lambda$), referred to as a hypercube.

Distortions introduced during endoscopic imaging (Fig. 2a) must be compensated in the hypercube to provide accurate spatial and spectral information. The intrinsically co-registered wide-field and line-scan HSI data were used to determine the geometric transformation matrices that are to be applied to reconstruct the hypercube. The full computational process used to achieve this is detailed in a flow chart in Supplementary Fig. 1. First, several pre-processing steps were applied to the wide-field images for correction of: white and dark background intensity; honeycomb structure arising from the imaging fibre bundle; and barrel distortion of the endoscopic images (see Methods). The wide-field image registration process estimates geometric image transformation matrices between each image to form a single panoramic image from multiple wide-field endoscopic images (Fig. 2b), which would be acquired during standard freehand endoscopic operation. The geometric image transformation matrix

for each image is the $3 \times 3$ matrix that includes all relative linear shift-invariant image transformations among wide-field images such as translation, rotation and magnification, identifying the endoscope imaging position and distance between the endoscope and the sample. The line-scan HSI (Fig. 2c) is then transformed onto a set of global spatial coordinates using these geometric transformation matrices thereby minimising the influence of distortions arising from freehand motion. This is possible because the optical design of the imaging system enables wide-field images and hyperspectral image to be intrinsically co-registered at every spatial position. The corrected line-scan hyperspectral image are then combined and overlapped areas (arising from the finite slit width) are averaged, resulting in the reconstruction of the final 3D hypercube (Fig. 2d).

**Simulation and initial experimental validation of HySE**. A computational simulation was performed to demonstrate the proof-of-concept of using wide-field imaging data for the hyper-cube reconstruction process. An in silico USAF 1951 test target was created with each pattern in the target randomly assigned one of six reflectance spectral profiles, sampled in 100 spectral channels (Supplementary Fig. 2). To emulate standard wide-field endoscopic imaging of the in silico target, a region of 512 by 512 pixels was cropped and all spectral information was integrated. Then, a circular mask of 250 pixels radius, including a honeycomb structure akin to that found in our imaging fibre bundle, was applied to the cropped images. Translation, rotation and magnification transformations were then applied to the simulated endoscopy images (Fig. 3a) to test the impact of these image distortions on wide-field image registration process. A total of 74 simulated wide-field endoscopic images were obtained by moving the region of the target by 8 pixels along the $x$ axis each time. Each wide-field endoscopic image was then given randomly selected $y$ axis movement (2–7 pixels), rotation (0.54–0.71°), magnification (−0.15 to 1.36%), and randomly assigned dark noise (approximately 2% of

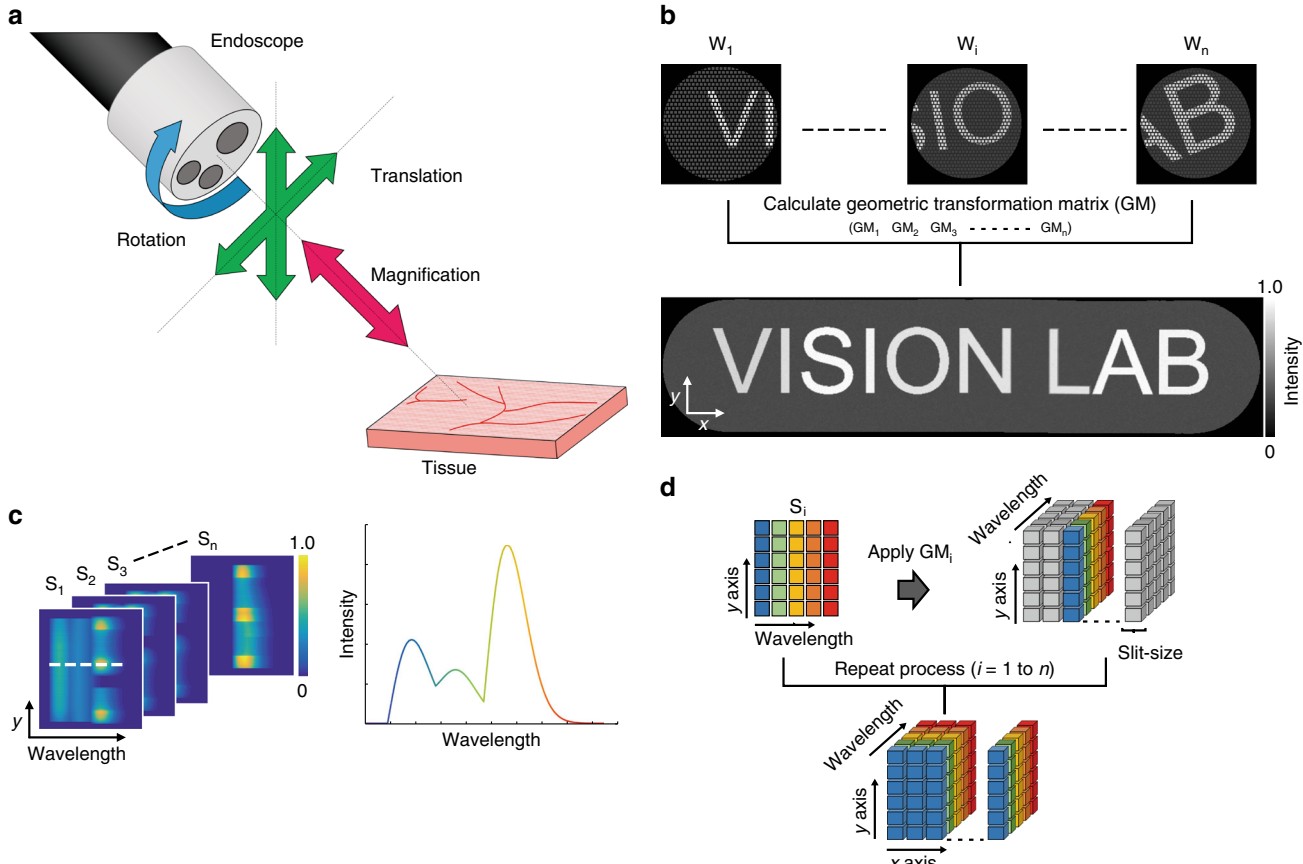

**Fig. 2** Schematic of the hypercube reconstruction process. **a** Three types of image deformation occur during flexible endoscopic imaging: translation, rotation and magnification. **b** To account for these deformations in composing the hypercube from the line-scan hyperspectral data (from sCam in Fig. 1), geometric transformation matrices ($GM_{1...n}$) are estimated using co-registration of the independently acquired wide-field images $w_{1...n}$ (from wCam in Fig. 1). Scale arrows are 60 pixels. **c** 2D spectral images ($S_{1...n}$) obtained at different positions contain spatial information on one axis ($y$) and spectral information on the second axis (wavelength). The spectrum encoded in the white dashed line on the image (left) is shown in the graph (right). **d** Applying the calculated geometric transformation matrices obtained in **b** to the spectral images **c** yields the reconstructed hypercube

maximum intensity) as image distortions, parameters determined based on known experimental conditions.

The geometric image transformation matrices of each image were computed and a wide-field panoramic image was then generated[43] (Fig. 3b and Supplementary Movie 1; see also Methods). Simulated line-scan hyperspectral image were created by selecting the central part of the in silico USAF target corresponding to a given slit width; intensity information was integrated along the slit ($x$ axis). The resulting 2D spectral image contained spatial ($y$) and spectral ($\lambda$) information as in our experimental implementation. The hypercube reconstruction process was performed as described above using simulated line-scan hyperspectral image based on the geometric transformation matrices determined from the simulated wide-field images. Representative slice images of the reconstructed hypercube at four simulated wavelengths (Fig. 3c) show correct spatial patterns with different spectral characteristics, i.e., features are absent from the in silico USAF target image when there is no spectral absorption defined at the simulated wavelength.

For experimental demonstration, a reflective USAF 1951 resolution target was imaged with an external illumination mode. A total of 360 wide-field and line-scan hyperspectral image were measured while moving the HySE tip along the $x$ axis at working distance of 5 mm using a motorised stage. The calculated geometric transformation matrices from the wide-field images produced a wide-field panoramic image (Fig. 3d) and were then

used to reconstruct the full 3D hypercube (slices from wavelengths at 495.7 and 610.4 nm shown in Fig. 3e). The USAF target hypercube was successfully reconstructed in both cases, which indicates that estimating geometric transformation matrices can effectively correct image distortion and retrieve global spatial coordinates for hypercube reconstruction.

**Technical evaluation of HySE performance.** As spatial and spectral resolution are intrinsically linked due to the line-scanning approach used in HySE, we first determined these parameters to ensure that adequate performance was achieved for clinical imaging applications. Spectral resolution was assessed by comparing our measured hyperspectral data to the calibration data of a reference light source (Supplementary Fig. 3a, b), whereas spatial resolution was determined using the contrast measured from a standard USAF 1951 test target using a motorised stage as above (see Methods).

First, the effect of the grating on spectral resolution and bandwidth of a single spectral image measured by HySE was tested. To obtain optimal spectral resolution, the entrance slit size was fixed at 10 μm and the reference light source, consisting of Mercury and Halogen-Argon lamps, was placed at the distal end of the endoscope. The spectral resolution and bandwidth obtained by using the three different gratings with 50, 150 and 300 lines/mm were $(2.85 \pm 0.39)$, $(1.10 \pm 0.01)$, $(0.46 \pm 0.05)$

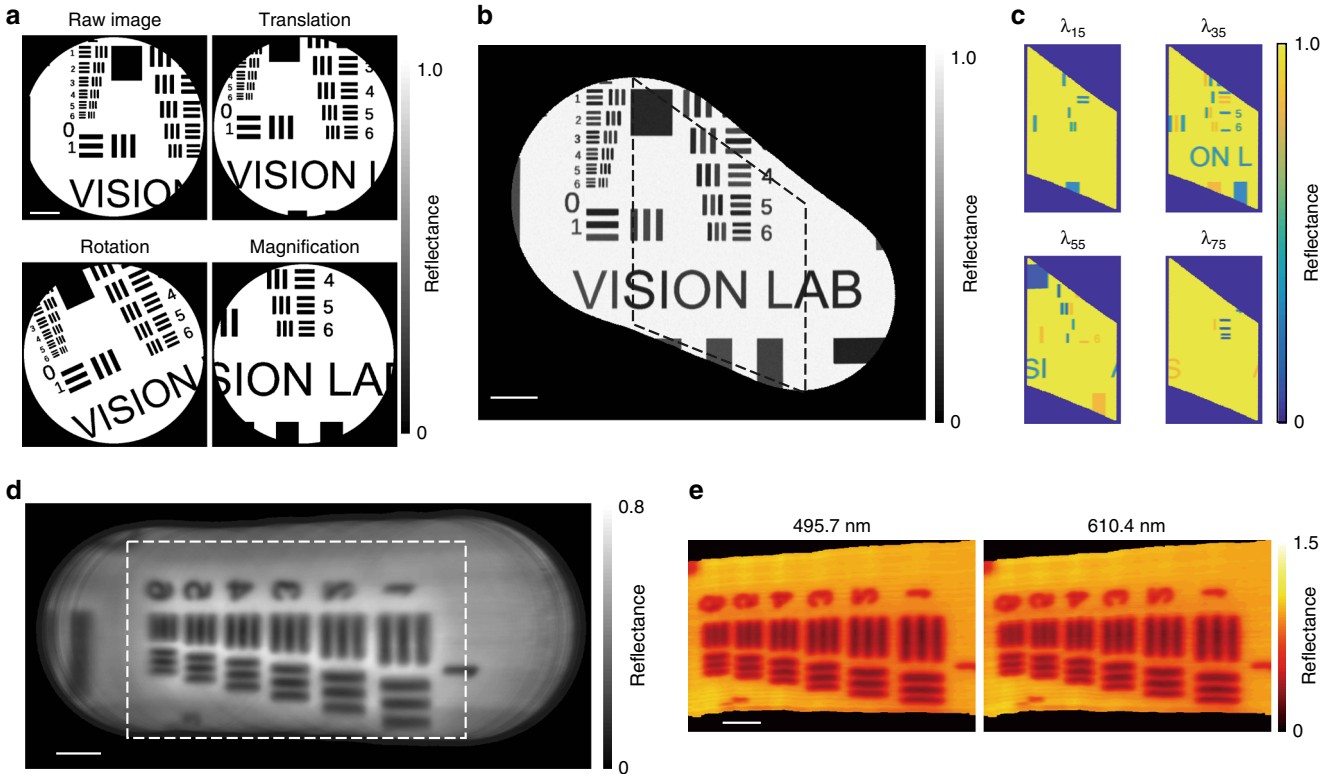

**Fig. 3** Hyperspectral endoscopic line-scan slices can be co-registered using wide-field reference images. **a** The three main types of image deformation were first emulated in simulation. Three types of image deformation were applied during an endoscopic procedure, in which the image x axis was scanned. **b** Seventy-four wide-field images were successfully co-registered and a combined image produced. **c** The geometric transformation matrix obtained from the wide-field image registration procedure was then applied to the simulated spectral image data. Images are shown from four different simulated wavelengths. **d** Combined image from 360 experimentally measured and co-registered wide-field images of a USAF 1951 resolution target. **e** Images from the experimentally reconstructed hypercube at two different wavelengths: 495.7 and 610.4 nm. Images acquired using the external illumination method. Scale bar in simulation **a**–**c** = 100 pixels. Scale bar in experimental data **d**, **e** = 1 mm. Dashed squares in **b** and **d** indicate the slit scanning areas where spectral images were obtained

and 750, 250, 125 nm, respectively, which are consistent with specifications provided by the manufacturer. This indicates that the HySE can fully exploit maximum performance of the spectrograph and a grating could be optimised depending on the desired compromise between spectral resolution and bandwidth in a given application.

Next, spectral and spatial resolution of the HySE according to the slit size were investigated. As expected, spectral resolution was found to degrade [from $(2.85 \pm 0.39)$ to $(17.83 \pm 0.22)$ nm at the 50 lines/mm grating, from $(1.10 \pm 0.01)$ to $(5.98 \pm 0.01)$ nm at the 150 lines/mm grating and from $(0.46 \pm 0.05)$ to $(2.98 \pm 0.01)$ nm at the 300 lines/mm grating] with increasing the slit size (from 10 to 28 μm) (Supplementary Fig. 3c). In line with our simulation (Fig. 4a), USAF elements become indistinguishable as the slit size increases (Fig. 4b). The optimal spatial resolution estimated with a slit size of 10 μm at a working distance of 5 mm was $(119 \pm 12)$ μm. The implication is that the slit size has to be carefully selected to optimise both spatial and spectral resolution, which are inversely proportional to slit size, as well as the optical throughput of the spectrograph, which is directly proportional to slit size (Supplementary Fig. 3c). As expected for line-scan HSI, selecting an appropriate slit width is crucial to maintain high image quality while minimising image acquisition time. We found a slit size of 18 μm could enable freehand operation, with sCam exposure time of 21 ms and spectral resolutions in the range $(0.89 \pm 0.02)$–$(9.18 \pm 0.08)$ nm (300–50 lines/mm grating respectively; Fig. 4c), which is more than sufficient for sampling the spectral reflectance properties expected from biological samples.

**Evaluation of freehand HySE operation**. Technical performance evaluation of the endoscope was performed using a rigid translation stage, however, for practical endoscopic surveillance, freehand operation is necessary. In order to enable HySE data acquisition in real-time, spectral images were stored in the internal memory of sCam during image acquisition while widefield images were directly stored in the hard drive disk, which maximises imaging speed by reducing data saving time of sCam. Using the internal memory of sCam was problematic for synchronisation between sCam and wCam due to inaccessible internal indexing of sCam; thus wide-field imaging was performed slightly faster than spectral imaging to minimise the imaging time difference between spectral and wide-field imaging. The optimised image acquisition rate of spectral and wide-field imaging were $(20.81 \pm 0.07)$ and $(37.09 \pm 0.49)$ fps with sCam and wCam exposure time of 21 and 25 ms, respectively, enabling realtime line-scan hyperspectral measurement.

To evaluate the potential for freehand operation, we printed a coloured vascular tree pattern (Fig. 4d) by mimicking those commonly used during system characterisation in endoscopy[40] and photoacoustic tomography[44]. The pattern was illuminated internally and imaged by freehand operation of HySE. Supplementary Movie 2 shows the entire process of wide-field endoscopic imaging, wide-field image registration and hypercube reconstruction. Endoscopic scanning by hand caused substantial changes in imaging position, magnification and rotation not encountered during the use of the motorised stage. Nonetheless, the geometric transformation matrices could successfully produce a panoramic wide-field image

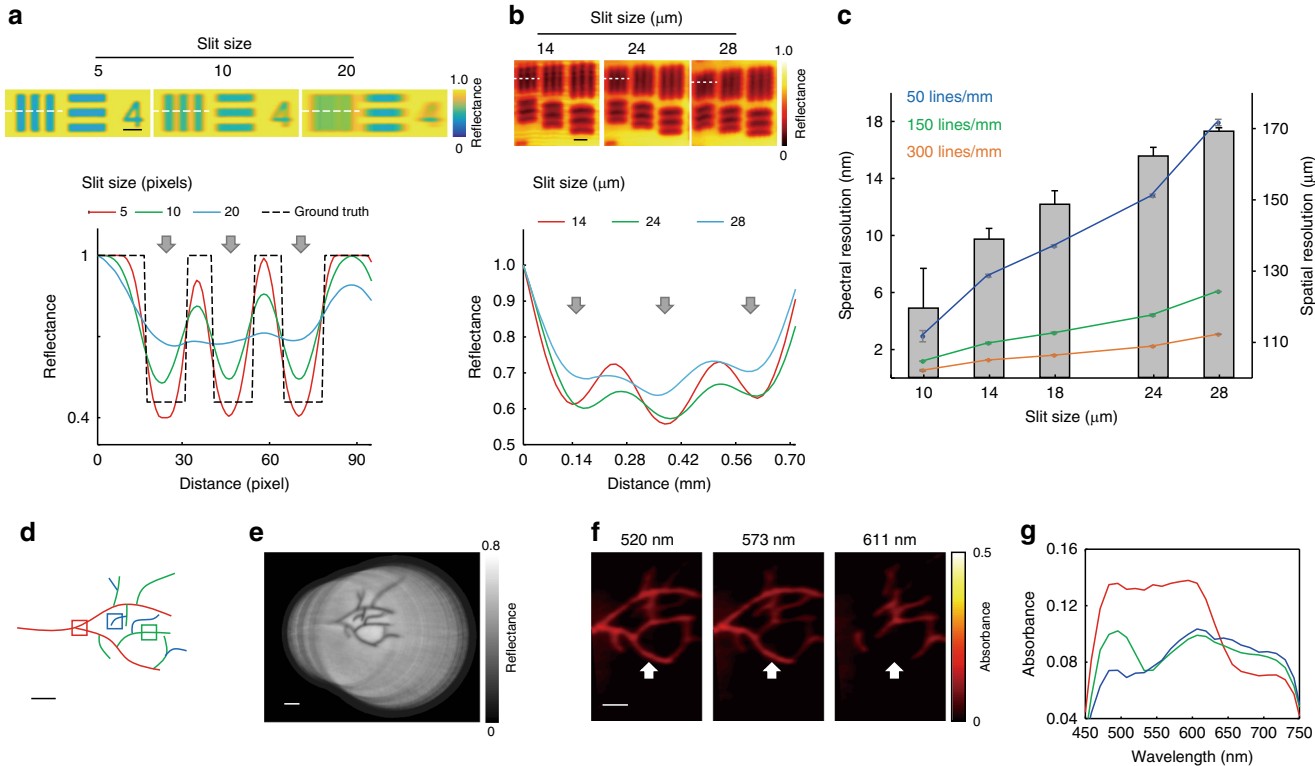

**Fig. 4** Characterisation of spatiospectral resolution of HySE. Spatiospectral resolution and signal intensity can be optimised by adjusting the spectrograph slit size, which enables hyperspectral imaging (HSI) during freehand motion. **a** Maximum intensity projection of simulated spectral images of a USAF 1951 test target at three different slit sizes (5, 10 and 20 pixels). Grey arrows indicate the three vertical lines across which the intensity profile (white dashed line) is quantified in the graph. The three chart elements are clearly resolved for slit sizes of 5 or 10 but at a slit size of 20 pixels, contrast is lost. **b** Experimentally measured images of a USAF 1951 test target at three different slit sizes (14, 24 and 28 μm). As in simulation, the three chart elements lose contrast as the slit size increases. Images show a representative slice of the hypercube at 518 nm. **c** The graph shows the trade-off between spatial resolution (bar graph; right axis) and spectral resolution (line graph; left axis) at the same exposure time for different slit sizes. Images acquired using external illumination. Error bars represent standard deviation in bars and lines. **d** A vascular tree phantom was created using three colours to demonstrate freehand HSI (see also Supplementary Movie 2). The internal illumination method was used during freehand imaging. **e** During freehand imaging, wide-field registration was performed. A combination of 86 endoscopic wide-field images is shown. **f** Representative slice images from the reconstructed hypercube are illustrated at three wavelengths. White arrows indicate the presence and absence of red vascular structures in the different single wavelength images. Colour bar indicates absorbance. **g** Absorbance was quantified within the red, green and blue squares shown in **d**. Images acquired using internal illumination. Scale bar in simulation **a** = 25 pixels, scale bar in experiment **b** = 500 μm and scale bars in **d-f** = 10 mm

(Fig. 4e) and reconstructed hypercube (single wavelength slices shown in Fig. 4f). As expected, the absorbance image at 611 nm clearly shows the disappearance of red coloured vascular patterns. Quantitative analysis of absorbance of three colour patterns as a function of wavelength also confirms this result (Fig. 4g). Next, the consistency of freehand HySE operation was evaluated by measuring relative length ratio and line profile (Supplementary Fig. 4a) of two branches from repeated experiments (Supplementary Fig. 4b, c). The coefficient of variation of the length ratio was 1.02% and the full width at half maximum of two peaks in line profile were $(13.4 \pm 2.4)$ and $(11.5 \pm 2.1)$ pixels, respectively, across repeated freehand experiments. These results indicate that freehand HySE operation and computational hypercube reconstruction can be performed with high repeatability. For ease of endoscope handling, the remaining measurements for this study apart from the ex vivo pig oesophagus imaging were acquired using a motorised stage, but the ability to accurately and consistently reconstruct both spatial and spectral data during freehand motion underpins the potential of the HySE for practical endoscopic surveillance.

**HySE enables reflectance colour separation.** To demonstrate the spectral fidelity of reflectance imaging via HySE, we measured a Macbeth Colour Checker chart consisting of 24 squares with

different colours (Fig. 5a). Qualitative examination of single wavelength images from the hypercube (Fig. 5b) indicates successful reconstruction and faithful colour estimation. Quantitative comparison of the spectral characteristics measured with HySE using three different gratings (50, 150 and 300 lines/mm) to the results obtained with a reference spectrometer shows excellent agreement (Fig. 5c). The mean square error and standard deviation for each wavelength are very low, at $(8.4 \pm 7.3) \times 10^{-4}$, $(8.5 \pm 4.2) \times 10^{-4}$, $(2.9 \pm 1.1) \times 10^{-3}$ and $(3.8 \pm 3.1) \times 10^{-3}$, respectively. The HySE can therefore measure a hyperspectral reflectance image with high colour accuracy based on the recorded spectral information regardless of line numbers in gratings.

**Tissue classification and oxygen saturation measurements.** In order to demonstrate the applicability of HySE in imaging biological tissue, we performed experiments using ex vivo cut chicken drumstick bone, which contained regions representing the skin and medullary cavity (Fig. 6a). Reconstructing the hypercube and calculating tissue absorbance showed clear qualitative differences between these two skin and bone tissue regions at different wavelengths (Fig. 6b). Unsupervised multivariate statistical analysis using a conventional principal component analysis (PCA) and $k$-means clustering ($k = 2$) algorithm[45]

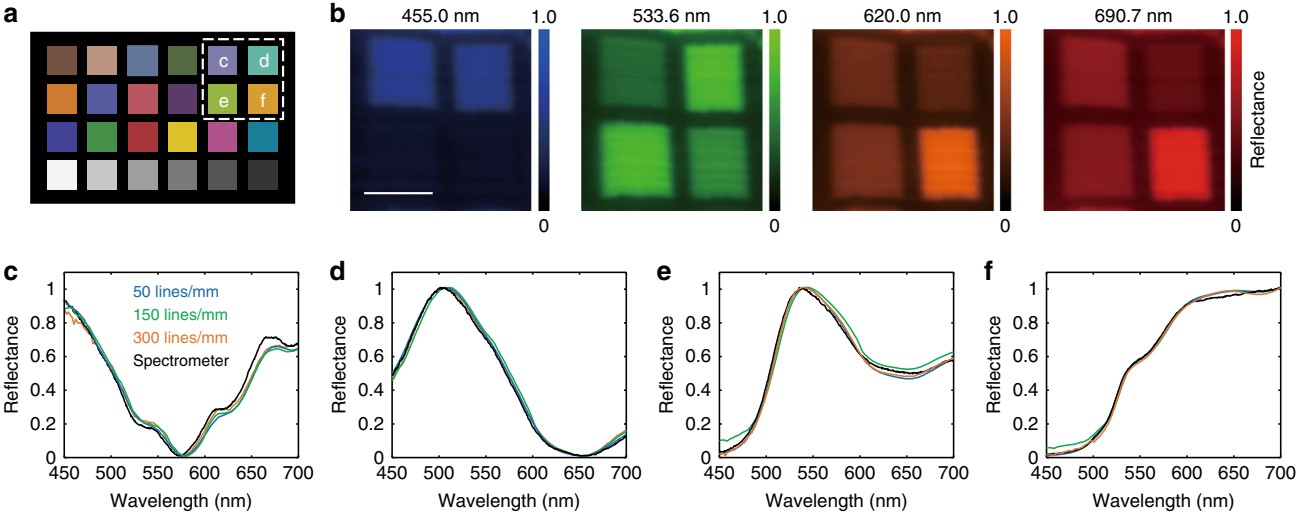

**Fig. 5** HySE exhibits excellent colour fidelity. **a** Schematic of colour samples interrogated in a Macbeth Colour Checker. **b** Representative slice images of the hypercube in the region of the white dashed line in **a** at four wavelengths. **c**–**f** Normalised reflectance intensity profiles of the four colour samples indicated by **c**–**f** in **a**. Black line indicates ground-truth reflectance signals recorded by a calibrated spectrometer. Blue, green and orange lines are reflectance signals in the hypercube measured by three different gratings (50, 150 and 300 lines/mm), respectively. Images acquired using external illumination. Scale bar = 1 mm

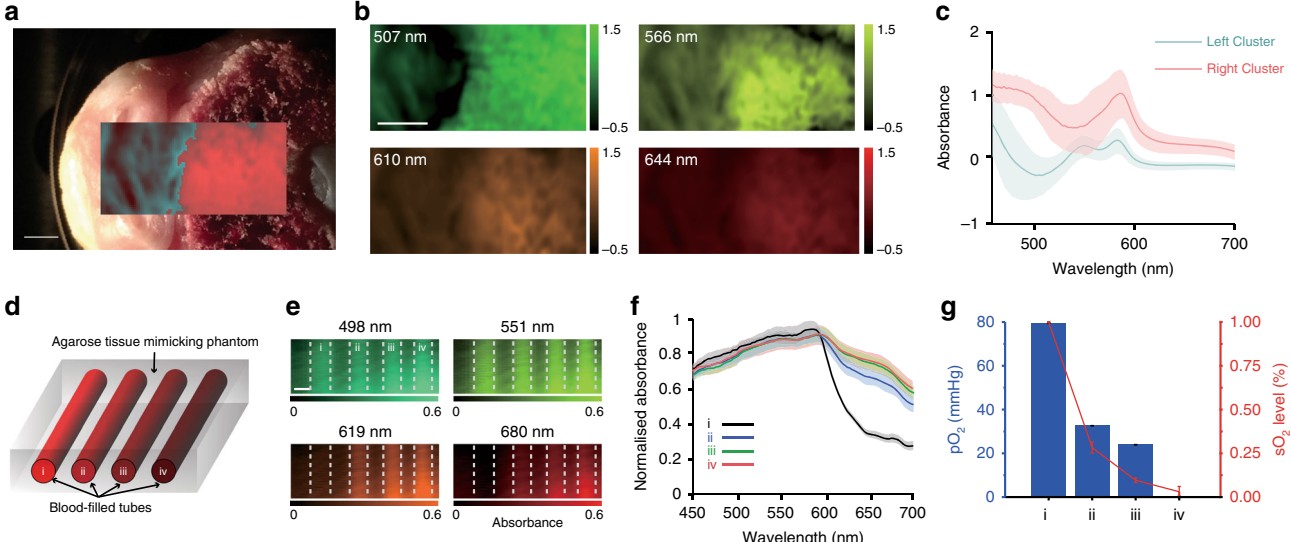

**Fig. 6** HySE enables quantification of physiological information. **a** Photograph of chicken bone tissue on a Petri dish. The overlaid false colour region indicates the hyperspectral imaging (HSI) area and shows the two tissue regions identified using principal component analysis and k-means (k = 2) clustering. **b** Representative slice images from the hypercube in the region of the dashed line in **a** at four wavelengths. The colour bars indicate relative absorbance. **c** Spectra of the identified clusters shown in **a**. **d** Schematic of tissue phantom with blood-filled tubes at four different oxygen levels indicated by (i)–(iv): (i) fully oxygenated blood; (ii, iii) partially oxygenated blood; (iv) fully deoxygenated blood. **e** Representative slice images from the hypercube at four wavelengths. Dashed line indicates tube boundaries, which are not absorbing. **f** Normalised absorbance spectra of the blood samples. **g** Quantitative comparison of partial oxygen pressure (pO$_2$) measured by a reference probe and oxygen saturation (sO$_2$) calculated by linear spectral unmixing of measured tube spectra. Solid lines and shaded areas in **c**, **f** indicate mean value and standard deviation of the absorbance profile, respectively. Error bars in **g** represent standard deviation in bars and lines. Images acquired using external illumination. Scale bars **a**, **b** = 2 mm. Scale bar **e** = 3 mm

accurately distinguished the spatial location of the two tissue regions (overlay on Fig. 6a). Different spectral features of absorbance in each cluster are apparent, with increased absorbance in the region of the bone (Fig. 6c).

Another key advantage of acquiring spectroscopy data in endoscopy is the ability to reveal the haemoglobin concentration and oxygen saturation, which are altered early in the development of cancer[46,47]. We therefore tested whether our HySE is able to discriminate different blood oxygen saturation levels. Four different blood samples at different oxygen saturations (fully oxygenated, two

intermediate levels, and fully deoxygenated) contained within plastic straws were embedded within a tissue mimicking phantom composed of agarose gel and intralipid (Fig. 6d, see Methods). HySE was scanned over the surface of the phantom and absorbance maps were calculated from the reconstructed hypercube (Fig. 6e). Fully and partially oxygenated blood samples have lower light absorbance over 620 nm, whereas the others have high absorbance (Fig. 6f), as would be expected from the literature[48]. Calculated sO$_2$ of the blood samples were (0.999 ± 0.002), (0.283 ± 0.032), (0.096 ± 0.013) and (0.030 ± 0.030)%, respectively (Fig. 6g). We validated the accuracy

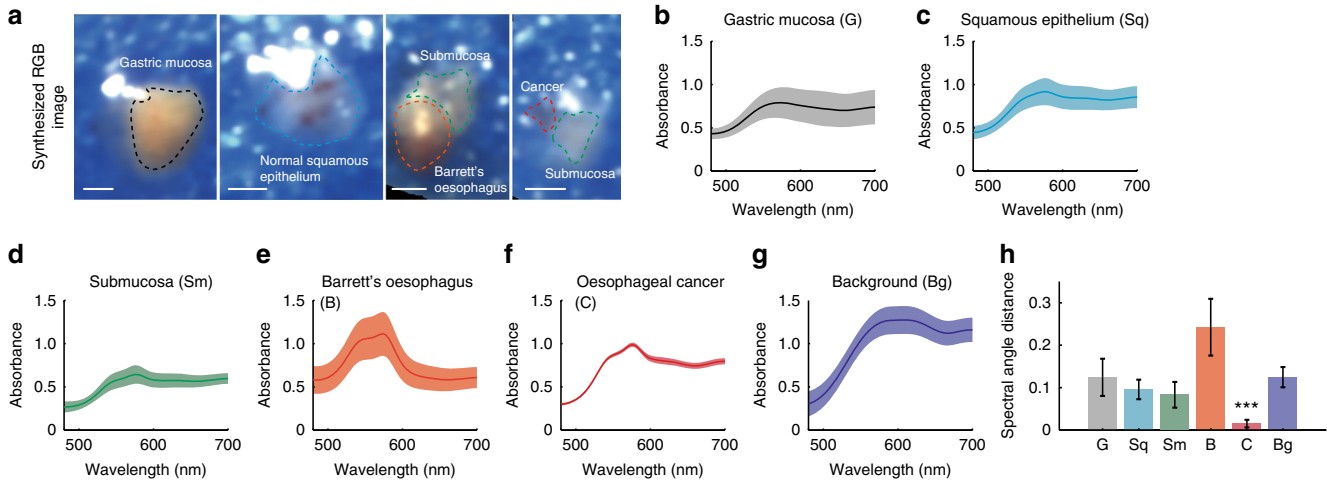

**Fig. 7** Hyperspectral imaging (HSI) of ex vivo human tissue from patients shows distinct spectral profiles. **a** Four representative synthesised RGB images of each sample. Dashed line indicates boundary of gastric mucosa, epithelium, submucosa, Barrett's oesophagus and cancer, respectively. **b–g** Spectra of the identified tissue types shown in **a**. Solid lines and shaded areas in **b–g** indicate mean value and standard deviation of the absorbance profile, respectively. **h** Spectral angle mapper analysis using average spectral profile of oesophageal cancer as a reference signal. Error bars represent standard deviation. *** indicates $p$-value < 0.001. Statistical analysis was performed using a one-way ANOVA with post-hoc tests. All tissue was measured using the external illumination method. Scale bars = 1 mm

of these sO$_2$ calculations against oximeter readings of partial oxygen pressure (pO$_2$): 79.54 ± 0.17, 32.62 ± 0.09, 23.92 ± 0.10 and 0.05 ± 0.02 mmHg, respectively. Some non-linearity exists between calculated sO$_2$ and pO$_2$ measured by the oximeter. This is most likely because pO$_2$ was directly measured in blood while sO$_2$ was calculated via the linear spectral unmixing method using experimentally measured oxy and deoxy absorption spectra in the tissue phantom as endmembers, which includes scattering and absorption effects caused by the phantom. Nonetheless, the overall trend of sO$_2$ and pO$_2$ values are consistent (Fig. 6g). These results indicate that HySE can delineate different tissue regions and have the capability of measuring blood oxygen levels based on hyperspectral information.

**Distinct spectral profiles of pathologic oesophageal tissue.** To demonstrate the potential for HySE to be applied in a clinical setting in the future, we measured ex vivo normal and pathologic human tissues ($n = 12$) collected from three patients: normal gastric mucosa, normal squamous epithelium, normal submucosa, Barrett's oesophagus and oesophageal cancer. Synthesised Red-Green-Blue (RGB) images were generated for each of the human tissue samples from the reconstructed HSI (Fig. 7a, Supplementary Fig. 7). The boundaries of each tissue type (dashed lines in synthesised RGB images) were selected based on histopathology analysis and the operating endoscopists (M.d.P. and W.J.) removing background areas and specular reflections. The average and standard deviation of absorption spectra in the selected areas of gastric mucosa, squamous epithelium, submucosa, Barrett's oesophagus, oesophageal cancer and background were extracted (Fig. 7b–g). Normal gastric mucosa, squamous epithelium, and submucosa appear to have similar average absorption spectra but different in scale, whereas Barrett's oesophagus and oesophageal cancer show distinct spectral profiles.

In order to analyse differences among the absorption spectra of each tissue, a spectral angle mapper (SAM) algorithm was used (see Methods). SAM is able to quantify similarity between target and reference spectral signals independent from their absorption values by calculating an angle between two spectral signals. The average spectra of oesophageal cancer was used as the reference spectral signal for SAM analysis. The calculated angles

between the reference and absorption spectral signals of gastric mucosa, squamous epithelium, submucosa, Barrett's oesophagus, oesophageal cancer and background are 0.11 ± 0.05, 0.08 ± 0.02, 0.06 ± 0.03, 0.21 ± 0.05, 0.01 ± 0.01 and 0.11 ± 0.02, respectively (Fig. 7h). Moreover, statistical analysis of the calculated angles was performed using a one-way analysis of variance (ANOVA) with post-hoc tests. The absorption spectra of oesophageal cancer is significantly different from the others ($p < 0.001$), indicating that HySE has potential to diagnose oesophageal cancer during endoscopy based on its spectral profile.

**Real-time HSI of an intact ex vivo pig oesophagus.** In order to demonstrate the capability of HySE in an endoscopic imaging application, HySE was operated to simulate clinical conditions while inserted through the accessory channel of a clinical gastroscope in an intact ex vivo pig oesophagus model (see Methods, Supplementary Fig. 8a, b). To introduce motion, the clinical gastroscope was bent laterally from the right to left sides of the lumen by the operator (Supplementary Movie 3). Such motion changes the imaging position of the HySE enabling the line-scan, and also examines different morphology of the oesophagus mimicking a real-clinical endoscopy examination.

HySE was operated with the internal illumination method, a 50 lines/mm grating, and sCam exposure time of 25 ms, which enabled real-time spectral and wide-field image acquisition (20.8 ± 0.1 fps) during the clinical gastroscope movements (Supplementary Movie 3). The bandwidth of a single spectral image was 750 nm but the analysed spectral range was limited to between 450 and 710 nm due to the limited illumination power of the light source at further wavelengths. An exemplary wide-field colour image and synthetic RGB image created from the measured hypercube (Fig. 8a, b) both clearly show the morphology of the pig oesophagus lumen. After this standard reflectance imaging, the lumen was stained by spraying methylene blue (MB) solution, a common clinical procedure. MB signals in the lumen were well differentiated by HySE (Fig. 8d, e) and the absorption spectral profile was completely changed in MB staining regions (Fig. 8c, f). The measured absorbance of the normal and MB-stained tissue via HySE was consistent with the ground-truth absorbance obtained by a spectrometer (Supplementary Fig. 8c–e). Importantly for demonstrating the potential of HySE application in

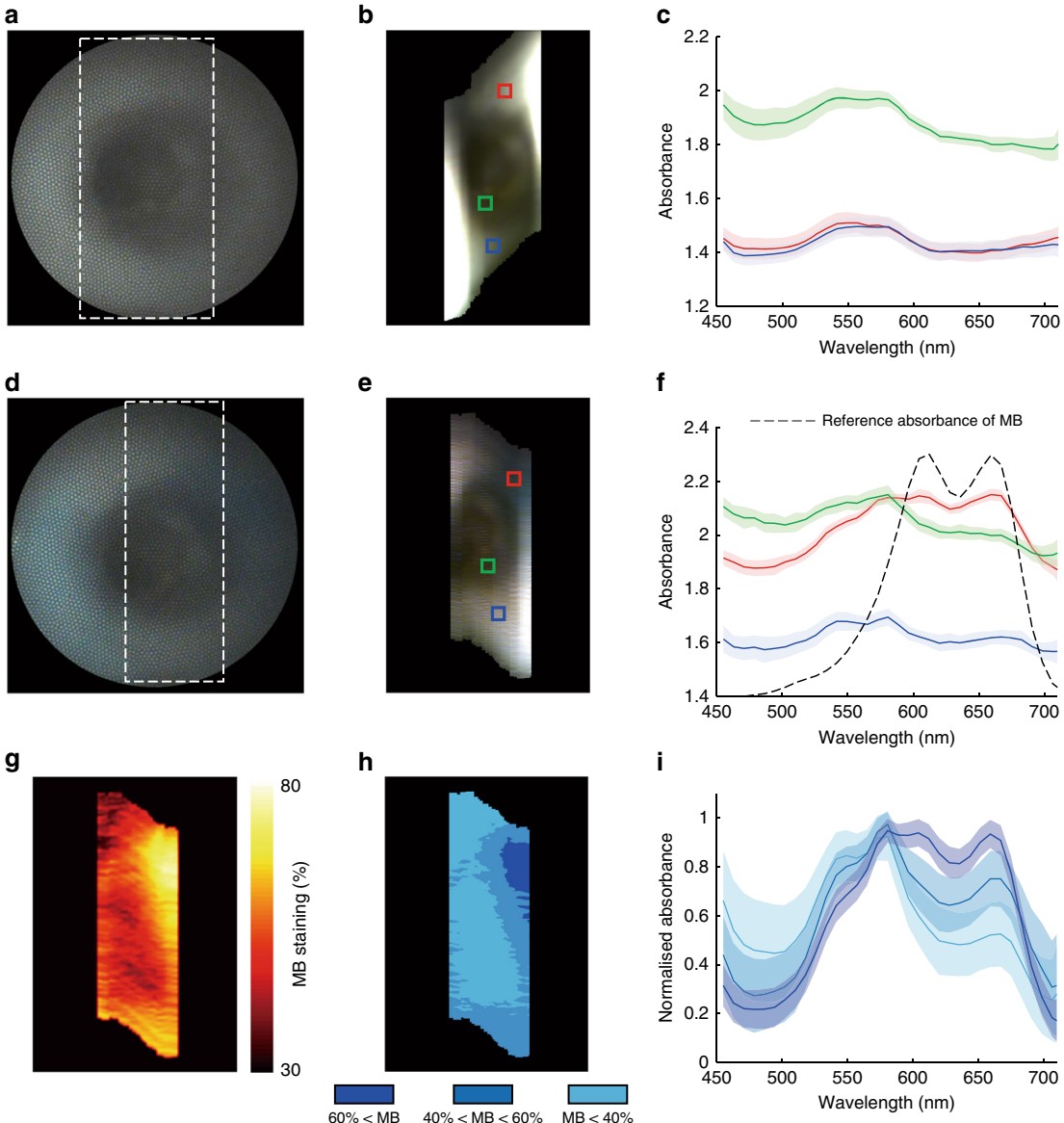

**Fig. 8** Hyperspectral imaging (HSI) of an intact ex vivo pig oesophagus. **a** Wide-field image, **b** synthesised RGB image and **c** absorbance spectra of the pig oesophagus. **d**–**f** Images and spectra from the same field of view after methylene blue (MB) staining. Dashed squares in **a**, **d** indicate the acquisition area of HSI. Colours in **c**, **f** correspond to the coloured squares drawn in **b**, **e**, respectively. **g** Concentration of MB staining estimated by linear spectral unmixing using the average absorption spectrum of normal pig oesophageal tissue and the experimentally measured absorption spectrum of MB as endmembers. **h** Segmentation of estimated MB concentration according to unmixing values. **i** Quantification of normalised absorption spectral profile of each segment in **h**. Solid lines and shaded areas in **c**, **f**, **i** indicate mean value and standard deviation of the absorbance profile, respectively. All hyperspectral images were measured using the internal illumination method

endoscopy, the relative concentration of MB staining could be assessed via linear spectral unmixing (Fig. 8g, h). The minimum–maximum normalised absorption spectral profile of each segment (Fig. 8h) was also quantified (Fig. 8i), indicating differences that are not apparent in the wide-field colour image. These results are encouraging, suggesting that HySE could be used in a real-clinical endoscopy examination.

## Discussion
HySE has the potential to improve diagnostic contrast for early cancer in the GI tract and lungs by enabling spectroscopy data to be recorded and displayed over the endoscope field-of-view. To overcome the practical data acquisition and reconstruction challenges associated with geometric distortions encountered during flexible endoscopy, our HySE approach measures both

line-scan hyperspectral and white light images simultaneously but does not require rigid mechanical scanning or any prior imaging information, using instead the co-registered white light images to correct for geometric distortions encountered during freehand endoscopy. Our concept was successfully demonstrated in simulations and experiments and satisfied the major requirements for HSI endoscopy by providing: high spatial resolution of up to 120 μm (comparable with previous studies of the same babyscope)[49]; high spectral resolution of up to 0.46 nm; accurate panoramic registration of freehand imaging data; real-time spectral (>20 fps) and wide-field (>35 fps) data acquisition and excellent colour fidelity. The imaging speed can be improved to over 29 fps by binning two vertical pixels. Supplementary Table 1 shows a summary of specifications of HySE compared to previously reported HSI endoscopy implementations, indicating that

the system satisfies the optical requirements for the real-time clinical HySE. We also demonstrated the potential for quantitative data extraction from biomedical samples. We first showed that unsupervised multivariate analysis can differentiate tissue regions in chicken tissue according to their spectral profile. We then established the capability of the system to measure oxygenation saturation levels in blood in controlled tissue mimicking phantoms, and derived absorption spectra from a variety of ex vivo tissue types in biopsies taken from the human oesophagus and stomach, showing clear differences between the spectra of oesophageal cancer and other tissue types. Finally, we demonstrated the potential of HySE in realistic endoscopic imaging scenario within an intact ex vivo pig oesophagus, obtaining relative concentrations of an applied stain using linear spectral unmixing.

There are several key advantages to this approach. First, the combined line-scan hyperspectral and wide-field imaging configuration exploits a computational wide-field image registration process that estimates relative geometric image transformations among measured wide-field images to replace rigid scanning, overcoming the traditional challenge of applying line-scanning HSI to a flexible clinical endoscope. HySE does not require any prior knowledge of imaging condition or reference images to create a panoramic image, thus is less vulnerable to image distortion caused by unknown endoscope and tissue movements and compatible with conventional WLE. Second, HySE enables real-time hyperspectral data acquisition, with the line-scan HSI acquisition rate of over 20 fps. In freehand HySE operation, a hypercube (75 spectral channels) containing over $6.64 \times 4.35$ cm area information can be fully acquired within 3.44 s. The area scanning, which sweeps out the final spatial ($x$) dimension of the hypercube, is acquired during normal freehand endoscope operation; it is not a controlled movement. Finally, the spectral resolution and bandwidth of a single spectroscopic image could be flexibly optimised according to application requirements. Our current spectrograph exploits three gratings (50, 150 and 300 lines/mm) that have optimal spectral resolution of $2.85 \pm 0.39$, $1.10 \pm 0.01$ and $0.46 \pm 0.05$ at bandwidths of 750, 250 and 125 nm, respectively, which would extend the applicability of HySE in practical clinical applications. This flexibility would be useful in combining studies of, for example, ultraviolet (UV) autofluorescence from molecules such as collagen and Nicotinamide adenine dinucleotide (NADH)[50] with contrast from haemoglobin absorption[48] and nuclear scattering[51–53]. Therefore, HySE provides a faster acquisition than rigid mechanical spatial scanning methods, tunable filter spectral scanning methods and provides richer and flexible spectral information compared with previous methods[15,54] (see quantitative comparison in Supplementary Table 1).

Nonetheless, for future clinical applications, HySE still requires several improvements. Wide-field images reveal imaging position and distortion information through the geometric transformation matrix, however, there may be difficulties in extracting features from the images if they are homogeneous or have low contrast. If this proved to be problematic, we could exploit label-free virtual chromoendoscopy approaches such as narrow band imaging[55,56] to improve tissue contrast. In addition, the spectral resolution of HySE is inversely proportional to the bandwidth of a single spectroscopic image because of the exploited gratings. In order to obtain a wide spectral range with high spectral resolution, a grating with large line numbers should be used and multiple spectral images measured with different grating angles (Supplementary Fig. 3d). This limitation can be easily solved by exploiting a camera with larger sensor size, or by using a priori knowledge of the target spectral features. Finally, HySE obtains HSI data ($y$, $\lambda$) in real-time but the hypercube is currently reconstructed offline, which limits practical clinical endoscopic

surveillance. The required time of data acquisition and saving of 59 line-scan hyperspectral image, data loading and pre-processing, wide-field image registration and reconstruction of a hypercube with 51 spectral channels are 1.99, 7.09, 1.86 and 37.24 s, respectively; thus offline post-processing for hypercube reconstruction requires approximately 50 s, but processing speed improvements could be achieved using C-based methods or parallelised computing architectures.

In spite of these limitations, HySE exhibits high spatial and spectral resolution as would be expected from a line-scan HSI approach but enables freehand operation and is built around a clinically applicable babyscope. Given the promise in our initial ex vivo human tissue and pig oesophagus experiments, our next steps are to expand these ex vivo trials and if results are validated in a larger number of human specimens progress to in vivo clinical trials. These trials will help us to establish whether the promising diagnostic potential of point spectroscopy reported in the GI tract can be exploited by the HySE system, opening up new era for clinical HSI.

## Methods

**HySE system**. The HySE (Fig. 1) was designed and built around a flexible clinically approved (CE marked) endoscope (Polyscope, PolyDiagnost) that consists of a disposable sterile catheter and a reusable imaging fibre bundle with 10,000 individual fibrelets. The disposable catheter contains an illumination fibre, an imaging channel, into which the imaging fibre bundle is inserted, and an accessory channel for introduction of instruments. The entire catheter system is designed to be inserted within the accessory channel of a full endoscope, in a so-called babyscope mode. The imaging system is located at the proximal end of the imaging fibre bundle and consists of an imaging spectrograph (IsoPlane 160, Princeton Instruments) that uses an electron multiplying CCD camera (sCam, ProEM 512, Princeton Instruments) as the detector array, and a monochromatic or colour complementary metal oxide semiconductor (CMOS) camera (wCam, GS3-U3-41C6m-C or GS3-U3-51S5C, Point Grey Research) depending on applications.

The proximal end of the fibre bundle was imaged and magnified by an infinity corrected objective lens (40×, NA 0.6, Nikon). The magnified image was split into two beams at a ratio of 1:9 using a non-polarising beam splitter cube (BS025, Thorlabs); 90% of the beam went to the spectrograph and the other 10% was measured with the wide-field camera. The tube lenses (L1, $f = 75$ and L2, $f = 150$ mm) positioned along the optical path of the spectrograph and wide-field camera, respectively, maximised the fit of the fibre bundle image to the sensor. For spectroscopic imaging, a mechanical slit with micrometre driven manually adjustable slit width and one of gratings (50 lines/mm with 500 nm blaze, 150 lines/mm with 500 nm blaze and 300 lines/mm with 500 nm blaze, Princeton Instruments) was used. The bandwidth of a single spectral image obtained by gratings with 50, 150 and 300 lines/mm were 750, 250 and 125 nm, respectively (Supplementary Fig. 3d). When gratings with 150 and 300 lines/mm were exploited, at least two or three spectral images with different centre wavelengths were needed to obtain a bandwidth over 300 nm. The 300 lines/mm grating was used for the experiments of system characterisation and ex vivo tissue measurements that requires high spectral resolution, without any requirement for high temporal resolution. The 50 lines/mm grating was used for real-time HSI applications (freehand imaging and ex vivo pig oesophagus measurements). The spectrograph operation was controlled by Light Field software v6.7 (Princeton Instruments). For experiments where quantitative control of endoscope movements with static working distance was required, a motorised translation stage was utilised (MTS50/M-Z8, Thorlabs). All equipment was synchronously controlled in Labview 2017 (National Instruments) environment.

Samples were illuminated using a white LED (UHP-T2-LED-White, Prizmatix) and halogen light source (OSL2, Thorlabs) with a Halogen light bulb (JCR 21-150W, KLS) whose emission spectra spanned across visible (400 nm) to NIR (750 nm) region. For internal illumination (through the illumination fibre of the catheter), the light from the white LED was focused onto the illumination fibre using an achromatic doublet lens ($f = 30$ mm) and an objective lens (60×, NA 0.9, Olympus). Although the internal illumination provided by the fibre gave static illumination pattern during scanning procedures, it delivered limited optical power to an area narrower than the imaging area as shown previously[57] due to limited numerical aperture and relative size of the illumination fibre in our operating conditions under test (2.1 cm in diameter at working distance of 2 mm); thus internal illumination was used for the experiments of freehand imaging and ex vivo pig oesophagus measurements to demonstrate practical applicability of the HySE in real-clinical applications. For external illumination (direct sample illumination), the light source was positioned 10 cm away from the sample. External illumination provides high power over a large exposure area (9.2 cm in diameter), hence was used for the majority experiments for system characterisations that requires long working distance for the large imaging area described in this study. The power of

illumination light of internal and external illumination at a sample position was $12.00 \pm 0.43$ and $22.63 \pm 0.02$ mW, respectively, as measured by using a calibrated thermal power metre (Laser Power Metres (PC-LINK), Laserpoint). Spectral calibration of the spectrograph was performed using a commercial software integrated spectral calibration algorithm and calibration reference light source (IntelliCal, Princeton Instrument). The curve fitting of measured spectral signals was performed using this proprietary software based on the Czerny–Turner model and the full width at half maximum of the curve fitting results is given as the spectral resolution of HySE.

**Image acquisition process**. Images were either acquired with translation motorised or freehand operation. In order to demonstrate the proof-of-concept of the proposed method, a series of wide-field and spectral images were acquired by scanning the endoscope using the motorised stage. To confirm that the same approach was feasible in freehand operation, imaging was also performed without the motorised translation stage. In this work, all figures except for freehand imaging and ex vivo pig oesophagus measurements were prepared based on data acquired using the motorised operation.

**Image pre-processing**. Before commencing image registration, four image pre-processing steps were performed on wide-field images (Supplementary Fig. 6). First, the recorded intensity of reflectance was normalised according to:

$$I_{nor} = \frac{(I - I_{dark})}{(I_{white} - I_{dark})}, \quad (1)$$

where $I_{nor}$ is the normalised intensity, $I$ is the measured intensity, and $I_{dark}$ is the dark counts measured from the sensor, and $I_{white}$ is the intensity measured from a standard white reflectance target. $I_{white}$ and $I_{dark}$ should be measured in every experiments because they are varied depending on experimental conditions. Second, the lens distortion caused by varying degrees of magnification along radial axis was corrected[58] according to:

$$x_c = x_0 + \alpha r \cos \theta, y_c = y_0 + \alpha r \sin \theta, \quad (2)$$

where $x_c$ and $y_c$ are corrected points, $x_0$ and $y_0$ are centre image positions, r is a radial distance from $(x_0, y_0)$ to $(x, y)$ in polar coordinate, θ is an angle between $x$ axis and line from $(x_0, y_0)$ to $(x, y)$, and $\alpha$ is a correcting coefficient. Third, honeycomb structures were removed using low-pass Fourier filtering, which cuts off high frequency components from the images including the first peaks arising due to the structure of the imaging fibre bundle[59]. Finally, the image was cropped to remove dark regions from outside the image field of the fibre bundle. The output from the pre-processing stage is referred to as the processed wide-field image.

**Wide-field image registration**. In order to extract features in the processed wide-field images, we used the Computer Vision Tool box in Matlab R2016b (Mathworks). The *Speed-Up Robust Feature Detection* (using detectSURFFeatures and extractFeatures function) was used to extract features in pre-processed images. During the wide-field image registration process, geometric transformation matrices were estimated (using estimateGeometricTransform function) by matching extracted features across several wide-field images, which contain information on the image distortion including translation, rotation, and magnification. The estimated geometric transformation matrices were then used for both a panoramic image generation from wide-field images and the hypercube reconstruction from spectral images (using imwrap function).

**Hypercube reconstruction process**. The 3D hypercube with dimensions of $(x, y, \lambda)$ was reconstructed by applying the calculated geometric transformation matrices to corresponding spectral images. For reconstruction of spatial information at specific wavelength, the corresponding columns of the spectral images were selected and horizontally repeated to make the dimension of the selected column match to the physical slit width. The resized column was transformed to its imaging position using the corresponding geometric transformation matrix. By placing all columns into their original imaging position and averaging overlapping areas, a wide-area spatial image at the specific wavelength can be reconstructed. Then, by repeating above process for all wavelengths, the 3D hypercube can be reconstructed from line-scan hyperspectral data.

**Equations for reflectance and absorbance calculation**. We used Eq. (1) for calculating reflectance[3]. In all cases, we calculated absorbance from the hypercube by following equation (Beer–Lambert Law):

$$A(x, y, \lambda) = -\log 10 \frac{I(x, y, \lambda)}{I_0(x, y, \lambda)}, \quad (3)$$

where $A(x, y, \lambda)$ is the calculated absorbance at the wavelength of $\lambda$ at the point $x$ and $y$ in the image and $I(x, y, \lambda)$ and $I_0(x, y, \lambda)$ are measured reflectance intensity from the sample and background at the point $x, y$ in the image, respectively[3,60]. Because measured signals were divided by reference background signals to obtain absorption values, the unit is a dimensionless arbitrary unit.

**Synthetic RGB image and binary mask generation**. For the visualisation, the hypercube can be converted to a synthetic RGB (colour) image using an artificial RGB filter (Supplementary Fig. 7). The spectrum of the RGB filter has Rayleigh probability density function (raylpdf function in Matlab) and centre wavelengths of each colour are 442, 518 and 579 nm, respectively. The amplitude of each filter was determined not to make a saturated synthetic RGB image. The hyperspectral information from each pixel is multiplied with the RGB filters, which determines R, G and B values of each pixel (Supplementary Fig. 7b). Synthesised RGB image is displayed using imshow function in Matlab (Supplementary Fig. 7c).

**Hyperspectral performance evaluation**. To evaluate the spatial resolution of the final hypercube, we imaged a USAF 1951 test target (#53-714, Edmund Optics) at a working distance of 5 mm and performed linear regression analysis of the measured contrast value and spatial frequency of defined USAF elements[49]. Imaging was performed using the motorised stage with a translation step size of 25 μm. Intensity profiles were extracted from the line pair targets and fitted to a sinusoidal function to extract the maximum and minimum values of the intensity modulation. After extracting the Michelson contrast, defined as:

$$\frac{I_{max} - I_{min}}{I_{max} + I_{min}}, \quad (4)$$

where $I_{max}$ and $I_{min}$ are maximum and minimum values of the intensity modulation, respectively, the spatial resolution was estimated using a Michelson contrast cut-off of 1% the minimum contrast.

Spectral resolution was then determined by exploiting a calibration light source with known spectral profiles (IntelliCal, Princeton Instrument). The reference light source consists of Mercury (250–550 nm) and Neon-Argon (585–965 nm) lamps that emit multiple peaks at specific wavelengths. The calibration source was placed at the distal end of the fibre and imaged via the HySE. Centre wavelength of the spectrograph and exposure time of sCam were set to 650 nm and 1 s, respectively. Five spectral images were measured by increasing a slit width and the full width at half maxima of each spectral profile was calculated to determine the spectral resolution of HySE system (Supplementary Fig. 3). We quantified the relative intensity measured by the spectrograph by integrating the intensity of all pixels in the endoscopic images measured at the same camera exposure time and light source power.

To evaluate the colour fidelity of the HySE, a Macbeth Colour Checker chart (ColorChecker Classic Mini, x-rite) was imaged at a working distance of 3 cm with a translation step size of 150 μm on the motorised stage using three different gratings (50, 150 and 300 lines/mm).

To evaluate the freehand imaging performance of the endoscope, a vascular tree phantom composed of three different colours was printed on a paper. Wide-field and spectral images were then acquired using a grating with 50 lines/mm while the endoscope was moved under freehand operation over the phantom over a range of working distances (2–3 mm).

**Chicken tissue preparation**. To test the ability of the HySE to delineate distinct tissue types, we used fresh chicken tissue, a food-grade chicken drumstick purchased from a local grocery market, as the test sample. An incision was made on the chicken tissue to exposure the hard tissue (bone) present in the chicken sample. The test sample was placed across a Petri dish to perform HySE imaging. The chicken tissue was measured at a working distance of 5 cm with a step size of 100 μm on the motorised stage. A total of 135 wide-field and spectral images were measured, giving a total scanning area was $4.5 \times 13.5$ mm, using exposure time of 500 ms. The sample preparation and measurement were carried out within 3 h to ensure freshness of the sample. PCA was performed using Matlab 2016R (Mathworks) under a graphics processing unit (GPU, GeForce GTX 1080, NVIDA) based computing platform. In order to fit within GPU memory, the matrix size of the hypercube was reduced by binning pixels in $5 \times 5$ spatial blocks. Singular value decomposition (using svd function in Matlab software) was performed to obtain principal components and then a number of significant principal components that represent over 99.9% of original data were selected (three principal components). Based on calculated principal components, all spatial pixel was subjected to binary classification using $K$-means clustering algorithm (using k-means function in Matlab software, $k = 2$).

**Tissue mimicking phantom preparation**. Tissue mimicking phantoms with defined optical properties that closely mimic biological tissue were fabricated using 1.5% agar as the base material. All chemicals were purchased from Sigma-Aldrich. In all, 0.75 g of agarose was dissolved in 48.5 mL of distilled water, and then was heated to the boiling point using a microwave oven[61]. The solution was left to cool to ~40 °C, and 0.1 mL of 20% intralipid was added to the solution and gently mixed to induce optical scattering. The phantom material was poured into a plastic sample holder and kept inside a refrigerator to set.

For oxygenation measurements, we used sealed transparent plastic straws filled with blood. Fresh heparinised mouse blood was collected from deceased mice at the Biological Resources Unit of the Cancer Research UK Cambridge Institute. In total, 4 mL of mouse blood was held separately in two 15 mL conical tubes. To one tube, 4 μL of 30% hydrogen peroxide was added and to the other, 6 mg of sodium hydrosulphite was added to make oxygenated and deoxygenated blood,

respectively[62]. To make blood samples with different oxygen concentrations, oxygenated and deoxygenated blood were rapidly mixed at ratios of 7:3 and 3:7. Partial oxygen pressure in blood was measured using a reference probe (OxiLite Pro, Oxford Optronix). Images of the prepared phantom were collected over an area of $2.2 \times 2.0$ cm (step size of the motorise stage is 200 μm) over a time period of 100 s (exposure time is 700 ms). Sample preparation and measurement were done within 60 min to minimise any biological variation in the blood oxygenation within the sealed tubes that may occur due to the presence of air bubbles in the straw.

We calculated oxygen saturation (sO$_2$) using linear spectral unmixing by following equation:

$$S(x, y, \lambda) = \alpha \times S_{oxy}(\lambda) + \beta \times S_{deoxy}(\lambda), \quad (5)$$

where $S(x, y, \lambda)$ is absorbance at the image point of $x$, $y$ and at the wavelength of $\lambda$. $S_{oxy}(\lambda)$ and $S_{deoxy}(\lambda)$ are absorbance of oxygenated and deoxygenated blood at the wavelength of $\lambda$. Coefficients $\alpha$ and $\beta$ are calculated by solving this equation with non-negative least squares algorithm. Then, sO$_2$ is calculated by following equations: $\alpha/(\alpha + \beta)$[63,64]. We used calculated average absorbance of blood samples (i) and (iv) as reference absorbance of oxygenated and deoxygenated blood, respectively.

**Human tissue preparation and SAM analysis.** Human tissues were collected at Addenbrooke's Hospital from patients ($n = 3$) undergoing diagnostic work-up or endoscopic therapy for Barrett's-related intramucosal oesophageal adenocarcinoma. The study received ethical approval by the Cambridgeshire 2 Research Ethics Committee (09/H0308/118) and informed consent was obtained from all patients. The targeted areas of normal (stomach cardia and oesophagus) or diseased mucosa (Barrett's oesophagus or early cancer) were collected using standard endoscopic forceps (Olympus), and endoscopic mucosal resection specimens were collected using a 2 mm diameter punch. All collected samples were positioned with the epithelial layer facing upward in individual containers together with soft sheets of sponge to minimise sample movement during transportation. Autoclaved phosphate-buffered saline was added to keep the samples hydrated during HSI. Images of the human tissue were collected over an area of $5 \times 5$ mm (step size of the motorised stage 50 μm) over a time period of 150 s (exposure time is 500 ms). Sample measurement was completed within 3 h from the biopsy to minimise any biological variation in tissue sample. Samples were then fixed and underwent standard histopathology analysis. All human tissue measurements were performed using the external illumination method and a grating with 300 lines/mm.

For quantitative comparison of spectral profiles of each tissue type, SAM was exploited[65]. The spectral angle, $\alpha$, was calculated by following equation:

$$\alpha = \cos^{-1}\left(\frac{\sum_{\lambda=1}^{n} t_\lambda r_\lambda}{\left(\sum_{\lambda=1}^{n} t_\lambda^2\right)^{0.5}\left(\sum_{\lambda=1}^{n} r_\lambda^2\right)^{0.5}}\right), \quad (6)$$

where $t_\lambda$ and $r_\lambda$ are values of target and reference spectral profiles at the wavelength of $\lambda$, respectively. $n$ indicates a total number of a spectral channel. The average spectral profile of adenocarcinoma was used as the reference spectral profile. Then spectral angles of each types of human tissue were calculated, and then statistical analysis was performed using the one-way ANOVA with post-hoc tests (Matlab 2016R, Mathworks).

**Measurement of an intact ex vivo pig oesophagus.** A fresh ex vivo pig oesophagus and stomach (Medical Meat Supplies) was used to closely mimic real imaging conditions within the hollow lumen of human oesophagus. All blood was drained from the pig oesophagus and stomach and the lumen was cleaned before experiments. The HySE was introduced to the lumen through the working channel of a clinical gastroscope (GIF-1T240, Olympus), as would be the case in clinical endoscopic imaging (Supplementary Fig. 8). The pig oesophagus and stomach were inflated by injecting air using the clinical gastroscope. HSI was performed using a grating with 50 lines/mm and exposure time of 25 ms, which enables wide-field and spectral image acquisition at over 20 fps. During HSI, the clinical gastroscope was laterally bent from right to left sides of the lumen to scan HySE across the lumen. The scanning angle and direction were determined according to the orientation of the clinical gastroscope. To test whether HySE can delineate specific molecules based on spectral profile, MB (319112, Sigma-Aldrich) was used. In all, 5 mg/mL of MB was sprayed inside the lumen of the pig oesophagus using a syringe, and then the HSI was performed at same experimental conditions.

**Software.** Matlab R2017b and imageJ were used for image processing. Lightfield v6.7 (Princeton Instrument) was used to control the spectrograph and EMCCD. Labview 2017 (National Instruments) was used for synchronised control of the wide-field camera, spectrograph and EMCCD, and motorised stage. OriginPro 2015 was used for statistical analysis and all data are quoted as mean ± standard deviation unless otherwise stated.

## Data availability

Supplementary information is available on the online version of the paper. The datasets generated during and/or analysed during the current study are available in the Apollo—University of Cambridge Repository, https://doi.org/10.17863/CAM.17479. All other data including raw data are available from the corresponding author upon reasonable request.

## Code availability

All custom data analysis code is available online at: https://doi.org/10.17863/CAM.17479.

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

## Acknowledgements

This work was funded by CRUK (C47594/A16267, C14303/A17197, C47594/A21102) and the EU FP7 agreement FP7-PEOPLE-2013-CIG-630729. A.S.L. thanks the EPSRC, George and Lillian Schiff Foundation and the Foundation Blanceflor for funding. R.C.F. acknowledges an NIHR Research Professorship (RG67258). We also would like to acknowledge the Evelyn Trust and the The Kathy Shaw Memorial Oesophageal Cancer Fund for their generosity in funding the CCRC endoscopy unit and the endoscopy equipment, as well as the Human Research Tissue Bank, which is supported by the National Institute Cambridge Biomedical Research Centre, from Addenbrooke's Hospital. Additional infrastructure support was provided from the CRUK funded Experimental Cancer Medicine Centre. We would like to thank the following individuals for experimental assistance: the CRUK CI BRU and Dr Laura Bollepalli for technical assistance in the completion of this study; Dr Maria O'Donovan (Cambridge University Hospital) for performing the histopathological analysis; Mrs Bincy Alias (MRC Cancer Unit) for sample collection and preparation; and the staff of the endoscopy unit of the Cambridge Clinical Research Centre (CCRC) for the help during the endoscopic procedures. We would also like to thank Prof. Daniel Elson for helpful discussions with regard to the use of the Polyscope catheter.

## Author contributions

J.Y. and S.E.B. conceived of the idea and directed the work. J.Y. designed the HySE system, performed the simulations and optical experiments (including freehand imaging) and processed the hypercube data. J.J., D.J.W. and G.S.D. assisted in the design of HySE system. J.J. and A.S.L. prepared tissue phantom experiments. M.dP., W.J. and R.C.F. enabled the human tissue experiments. All authors wrote and revised the manuscript.

## Additional information

**Competing interests:** The authors declare no competing interests.

