## [Peer Review File · Nature Communications]

Reviewers' comments:

Reviewer #1 (Remarks to the Author):

This manuscript presents a new hyperspectral imaging endoscope that uses a white-light, full-field image that is co-collected with a spectral line (1 spatial dimension, 1 spectral dimension) to register position and to calculate-and-remove spatial distortion in the imaging optics. This system is small, flexible, contains no moving parts, and is integrated into a clinical-grade babyoscope.

The article is clear, well written, and shows promising results. That being said, it is not clear that this device in its present form is a superior development over what already exists. For example, the system by Kester, et al. [15] shows very similar resolution – slightly better spatial resolution, slightly worse spectral resolution. To be fair, this publication demonstrates a system with 512 spectral pixels (0.234 nm /pixel), but the effective spectral resolution is ~3.6 nm; thus, this system is greatly over-sampled (Kester has ~4-10 nm resolution). Further, the system by Kester acquires and processes images at 5.2 fps with biological samples; whereas, this system only achieves near those speeds with reflective AF targets and CCD binning. With biological samples, the exposure time (of the system in this manuscript) is at least 33x longer and requires external illumination. With internal illumination, this system would require at least 6x longer and would lose field-of-view. Another hyperspectral system, by Fawzy, et al. (not referenced, *BioMed Opt Exp* 6, p 2980 (2015)), also shows comparable acquisition rates, but requires just a few seconds for analysis.

Certainly, the system in this manuscript is unique and noteworthy. With grating modifications, as mentioned in the discussion, the spectrometer resolution could more closely align with the system-supported resolution, enabling a much broader band to be analysed at once – something other architectures cannot do. Additionally, improved computational power would undoubtedly improve performance. As it stand, though, the current generation of the system is too slow for practical (and clinical) hyperspectral imaging at >500 ms/spectral line.

A few questions that should be address:

- * Why the discrepancy/non-linearity between blood oxygenation and partial pressure in Fig 6G?
- * What is the white-dashed box in Figure 3B? Why is there blurring/distortion within this box and not outside of it?
- * "...Barrett's oesophagus and oesophageal cancer show distinct spectral profiles..." – you may need a different graphical representation or a metric (e.g., spectral angle distance [SAD]) to quantify or present this as it's not obvious – they look like scaled-shifted versions of each other. Further, the analysis of the data in Fig. 7 is extremely limited for such an important result.
- * Page 11, line 279, what is a "bespoke movement" mean?

In summary, I think this system has a lot of potential and is certainly a uniquely engineered system. I do not, however, believe the state of this system is so dominant over existing technologies to warrant its publication in *Nature Communications*. In its current form, I believe journals that the other HSI endoscopes have been published in would be more appropriate. I do believe, though, that if the authors demonstrated unprecedented bandwidth (e.g., a 50/mm grating, which is available for the IsoPlane 160, would provide ~800 nm bandwidth with the ProEM 512 CCD), improved the internal illumination to get down to near video-rate (at least for acquisition), and demonstrated this with the given bio-samples; this paper would merit publication in this particular journal.

Reviewer #2 (Remarks to the Author):

In their manuscript the authors report on the development of an endoscopic imaging system for hyperspectral imaging endoscopy for compensating motion of the operator during imaging sessions.

While hyperspectral imaging endoscopy is very well established for biological or clinical imaging, this work is based on the idea that collection of white light images obtained concomitantly with spectral measurements can be used to correct for free-hand induced motion artifacts (here authors focused on hand induced motion), using a previously developed scale- and rotation-invariant interest point detector and descriptor algorithm similar to SIFT but computationally faster. Transformation matrices are obtained using the white light information with one modality and applied then to the second modality (spectral). The approach works if enough features are present in the first registration modality and if motion occurs on a time scale which is smaller than the acquisition frame (obviously snr has to be large also).

Similar or alternative approaches have been used with the same or different imaging modalities before. And while the authors used a specific correction algorithm others are feasible also.

Overall I found the work interesting as an imaging paper, but I'm not sure the manuscript introduces substantial technical or method novelties enough for a journal such as Nature Communications.

Additional work also should be performed in my opinion on animal models in an in vivo setting where physiological motion is also present.

Reviewer #3 (Remarks to the Author):

The authors described a real-time hyperspectral endoscope they have developed that is claimed to overcome the present challenges of slow image acquisition and distortions during free-hand operation. Such an instrument would bring a new capability to medical screening and diagnostics since it could capture chemical signatures embedded in the reflection spectrum, or fluorescence spectrum, that are biomarkers of disease.

p3 line 81. please change "imaging mapping" to "image mapping"

p3 line 95 please change to "CE-marked"?

p13, line 326. For these simulations, just three wide bandwidth spectral images were employed. It is unclear if this provides enough wavelengths and enough spectral resolution to give a prediction for later development as a full hsi endoscope. Please provide planned path for testing at high spectral resolution and describe how the high resolution wavelength axis used in later simulations was created. What kind of calibration source was used. Linear or poly model fitted to recorded spectrum?

p13, line 337. How will limited internal illumination be overcome in a prototype, where the target is out of range of external illumination?

p14, line 360. How are radial distances known during rigid or freehand imaging, and can the tolerance for error be described? Was it possible to get a correspondence between radial distance and the magnification in the geometric transformation matrix?

p17, line 420. This describes a freehand imaging evaluation using a planar phantom drawn on paper. The critical problem is to be able to collect and render an hsi image from intravital space, usually convoluted and surrounding the endoscope. Thus it is not clear how well the authors' evaluation can predict performance under actual circumstances. Tissue phantoms and samples described later give a better prediction of the method of hsi endoscopy, yet this appears to still be an external imaging setup with the external illuminator. This reviewer believes additional tests of the endoscope using an animal are needed in order to justify a move to a clinical setting. Suggest working in the rat, using the GI surface or folds in the mouth area. Do the tissue reflectance spectra obtained compare well to known spectra, and are the geometric and light shading

corrections adequate?

Fig 1. The caption describes an internal fiber illuminator however also calls out either external or internal illumination. The external method is illustrated in the figure. Most of the simulation used external fiber illumination, however the two cases would very likely need different light/dark calibration. Please make sure the illumination method is given for each of the system tests.

Fig. 2. In B, only translation appears, would it be better to include rotation and magnification examples in the part? In all, the hsi image generation appears to be similar to pushbroom scanning if translation is the only variation. It isn't clear here how rotation and magnification are built into the geo-correction process. Apparently it would require the 2-D reference image.

Fig. 3. In A, it shows one acquired image and three variants made from the first, is this correct?. B is presumably a co-registered image where the rectangle covers the shared, or common, areas from all of the input images. C, missing letters result from spectral differences, but this should be reiterated in the text. Unclear how a.u units are established in the scale bars. D. Not clear what is different about the area enclosed by rectangle, more explanation of this process is needed. E. the images red and blue appear to be identical in the figure. Is that correct?

Fig. 6. A. Could not see the white dashed line in the submission. B. These images look promising. Please give method for a.u. determinations in B and C. D. change "blood filled" to "blood-filled".

Fig S2. B. Have wavelength values been associated with the simulation band numbers in x axis? If so please provide and describe the calibration.

Fig S6. Method for generating reflectance scale bars was not stated. How is the value of the reflectance determined?

Other. The instrument description seems to show that an OEM or custom built spectrograph system was used. Authors mention checking the spectral resolution with a calibration data from a reference source. Please specify the spectral properties of the calibration source, i.e., were Xe or Ar gas calibration lamps used which provide narrow spectral lines at different wavelengths? The method for determining the hsi wavelength axis should also be provided, even if a reference to these details was given. In certain places there are scale bars noting optical quantities such as a.u. etc. Please give more detail how these were arrived at.

Reviewer #1 (Remarks to the Author):

This manuscript presents a new hyperspectral imaging endoscope that uses a white-light, full-field image that is co-collected with a spectral line (1 spatial dimension, 1 spectral dimension) to register position and to calculate-and-remove spatial distortion in the imaging optics. This system is small, flexible, contains no moving parts, and is integrated into a clinical-grade babyscope.

The article is clear, well written, and shows promising results.

We would like to thank the reviewer for their time in carefully considering our manuscript. We hope that the changes made to the manuscript in this revision satisfy the concerns raised, particularly with regard to: the comparison of performance with existing systems; the need for improved speed of operation and spectral bandwidth; and further analysis of the human tissue data presented.

That being said, it is not clear that this device in its present form is a superior development over what already exists. For example, the system by Kester, et al. [15] shows very similar resolution – slightly better spatial resolution, slightly worse spectral resolution. To be fair, this publication demonstrates a system with 512 spectral pixels (0.234 nm /pixel), but the effective spectral resolution is ~3.6 nm; thus, this system is greatly over-sampled (Kester has ~4-10 nm resolution).

Further, the system by Kester acquires and processes images at 5.2 fps with biological samples; whereas, this system only achieves near those speeds with reflective AF targets and CCD binning.

With biological samples, the exposure time (of the system in this manuscript) is at least 33x longer and requires external illumination. With internal illumination, this system would require at least 6x longer and would lose field-of-view.

Another hyperspectral system, by Fawzy, et al. (not referenced, BioMed Opt Exp 6, p 2980 (2015)), also shows comparable acquisition rates, but requires just a few seconds for analysis.

We apologise for not clearly outlining the overall performance of our system or comparing fairly with other previous publications. We have made several amendments to our manuscript, which we hope address this point. They are divided into three main areas:

1. Results: Spectral / Spatial Resolution

According with this comment and the final comment of the reviewer regarding bandwidth, we have repeated our measurements using 3 different gratings (50 lines/mm, 150 lines/mm, and 300 lines/mm) and compared their performance using a single slit width of 10 μ m. The optimal spectral resolution of a single spectral image measured using three gratings is: 2.85 ± 0.39 , 1.10 ± 0.01 , and 0.46 ± 0.05 nm, while the pixel resolution is 1.465, 0.488, and 0.244 nm, respectively. These measurements indicate that HySE is only 2-fold oversampled, rather than 10-fold as indicated above. The Results have been updated to reflect this (p5, p8; Supplementary Figure 3c,d).

2. Results: Speed of Operation

To improve the speed of operation, we have now exploited the internal memory of sCam to avoid the long data transfer times that were limiting speed. Spectral images are now stored in the internal memory of sCAM during image acquisition process and then transferred to the hard drive disk of the computer. We have estimated the image acquisition and data saving time separately in the Table below. The findings indicate that horizontal pixel binning does not affect image acquisition speed but data saving time is improved due to the reduced file size. Spectral imaging without horizontal pixel binning but with 2 vertical pixel binning achieved imaging speed over 20 fps and 29 fps, respectively. Thus the

HySE now enables real-time spectral and wide-field image acquisition without losing spectral information.

	Fps	Saving time (ms)	File size (MB)
Full pixels	20.4 ± 0.5	6636.3 ± 173.9	125
Binning V1 H5	20.8 ± 0.1	1917.0 ± 127.3	24.9
Binning V1 H10	20.8 ± 0.1	1277.7 ± 163.3	12.4
Binning V2 H1	29.5 ± 0.2	3601.0 ± 318.5	62.5
Binning V2 H5	29.6 ± 0.2	1354.0 ± 374.2	12.4
Binning V2 H10	29.7 ± 0.1	889.0 ± 69.0	6.25

In reference to the comment regarding the illumination, we have optimised the optical setup for the internal illumination method and its optical power is now 12.00 ± 0.43 mW. However, as has been demonstrated previously by our laboratory, the illumination cone of the lighting fibre is narrower than the imaging cone of the imaging fibre bundle (Luthman *et al* 2018 J Biomed Opt 24, 3, 1-14). Therefore for characterization experiments we chose to use an external illumination that would uniformly illuminate the entire imaging area. Nonetheless, for freehand operation experiments, using the vascular phantom and in our new experiment within a pig oesophagus, we used the internal illumination and demonstrated operation at over 20 fps (Fig 4, 8), which we believe means that HySE could be practically used in clinical applications.

3. Discussion: Direct comparison of performance

We have now included a summary table of HySE performance compared directly to the two nearest previous publications (Supplementary Table 1). HySE measures real-time white-light colour images similar to the standard clinical endoscopy and additionally obtains line-scan spectral images. As demonstrated in the Table, HySE shows significantly improved spectral resolution, channel numbers, and imaging speed compared to the other systems.

Certainly, the system in this manuscript is unique and noteworthy. With grating modifications, as mentioned in the discussion, the spectrometer resolution could more closely align with the system-supported resolution, enabling a much broader band to be analysed at once – something other architectures cannot do. Additionally, improved computational power would undoubtedly improve performance. As it stand, though, the current generation of the system is too slow for practical (and clinical) hyperspectral imaging at >500 ms/spectral line.

Thanks to the suggestions of the reviewer, we have now demonstrated that HySE enables spectral and white-light colour image acquisition within the intact pig oesophagus with imaging speed of over 20 fps (Fig 8). We would believe the HySE has significant technical improvements compared previous hyperspectral endoscopy systems and holds potential for practical clinical applications.

A few questions that should be address:

* Why the discrepancy/non-linearity between blood oxygenation and partial pressure in Fig 6G?

We believe the origin of this non-linearity is the scattering in the tissue-mimicking phantom. Partial oxygen pressure was directly measured in blood but blood oxygen levels in the straws were estimated by the linear spectral unmixing method using experimentally measured oxy- and deoxy-haemoglobin absorption spectra in the tissue phantom as endmembers. We have now discussed this issue in the revised manuscript (p10, p11).

* What is the white-dashed box in Figure 3B? Why is there blurring/distortion within this box and not outside of it?

The white-dashed box indicates slit-scanning areas for hyperspectral imaging. We have now improved accuracy of wide-field image registration process thus the revised figure shows little blurring or distortion in the registered panoramic image. We have added a note to the legend of Figure 3 to explain the dashed box.

* "...Barrett's oesophagus and oesophageal cancer show distinct spectral profiles..." – you may need a different graphical representation or a metric (e.g., spectral angle distance [SAD]) to quantify or present this as it's not obvious – they look like scaled-shifted versions of each other. Further, the analysis of the data in Fig. 7 is extremely limited for such an important result.

We appreciate the suggestion of the reviewer to perform further analysis on these data. As the reviewer suggested, we have now calculated spectral angle distance using a spectral angle mapper (SAM) algorithm. In order to test whether spectral profile of oesophageal cancer is different from other tissue types, the average absorption spectral profile of cancer was used as the reference signal. We have updated Figure 7 (Fig. 7h) to show that there are differences in spectral angles among cancer and other tissues. We further performed statistical analysis of the results using a one-way ANOVA with post-hoc tests and found that the spectral profile of cancer is significantly different from the others (p -value < 0.001); therefore the HySE has potentials of detecting oesophageal cancer during hyperspectral endoscopy.

* Page 11, line 279, what is a "bespoke movement" mean?

This was intended to state that the movement was not controlled, so the sentence has been updated to better reflect this.

In summary, I think this system has a lot of potential and is certainly a uniquely engineered system. I do not, however, believe the state of this system is so dominant over existing technologies to warrant its publication in Nature Communications. In its current form, I believe journals that the other HSI endoscopes have been published in would be more appropriate. I do believe, though, that if the authors demonstrated unprecedented bandwidth (e.g., a 50/mm grating, which is available for the IsoPlane 160, would provide ~800 nm bandwidth with the ProEM 512 CCD), improved the internal illumination to get down to near video-rate (at least for acquisition), and demonstrated this with the given bio-samples; this paper would merit publication in this particular journal.

As suggested by the reviewer, we have upgraded HySE to include a 50 lines/mm grating and compared the results with 100 lines/mm and 300 lines/mm. The result is that HySE now measures a spectral image with a bandwidth of 750 nm at spectral resolution of 2.9 ± 0.4 nm using the 50 lines/mm grating. We have also performed hyperspectral imaging within the lumen of a pig oesophagus at a spectral imaging speed over 20 fps using the internal illumination method (Supplementary Movie 3).

Overall, we believe that the improved speed, registration accuracy and bandwidth of the upgraded HySE system show significant technical improvements against previous hyperspectral imaging systems and increase the novelty sufficiently to merit publication in Nature Communications. We would again like to thank the reviewer for suggesting these changes.

Reviewer #2 (Remarks to the Author):

In their manuscript the authors report on the development of an endoscopic imaging system for hyperspectral imaging endoscopy for compensating motion of the operator during imaging sessions.

While hyperspectral imaging endoscopy is very well established for biological or clinical imaging, this work is based on the idea that collection of white light images obtained concomitantly with spectral measurements can be used to correct for free-hand induced motion artifacts (here authors focused on hand induced motion), using a previously developed scale- and rotation-invariant interest point detector and descriptor algorithm similar to SIFT but computationally faster. Transformation matrices are obtained using the white light information with one modality and applied then to the second modality (spectral). The approach works if enough features are present in the first registration modality and if motion occurs on a time scale which is smaller than the acquisition frame (obviously snr has to be large also).

We would like to take this opportunity to thank the reviewer for their time in reading the manuscript. Based on the comments of the reviewers, we have now substantially revised and expanded the manuscript to: increase the speed of operation; improve the accuracy of wide-field registration; compare multiple gratings to achieve a bandwidth of 750 nm; include experiments performed in a clinically realistic setting using a pig oesophagus; and summarised technical characteristics compared with the nearest state-of-the-art. We hope that these changes will allay the concerns of the reviewer regarding the technical novelty of our work compared to others.

Similar or alternative approaches have been used with the same or different imaging modalities before.

The nearest hyperspectral imaging endoscopy approaches that we are aware of are those of Kester *et al* (2011, ref 15) and Fawzy et al (2015, ref 54). We have now included a table (Supplementary Table 1) to compare the performance of our system against these existing state-of-the-art. As demonstrated in the Table, HySE shows significantly improved spectral resolution, channel numbers, and imaging speed compared to these other systems.

And while the authors used a specific correction algorithm others are feasible also.

Indeed, other correction algorithms could also be employed for wide-field image registration, however, the existing approach already achieves competitive spatial resolution and with the improvements added in the revised manuscript suffers from minimal blurring or distortion. Therefore, for our purposes, the existing methodology appears to perform well. Nonetheless, if desired by the reviewer, we could compare this algorithm to others that they would be interested in.

Overall I found the work interesting as an imaging paper, but I'm not sure the manuscript introduces substantial technical or method novelties enough for a journal such as Nature Communications. Additional work also should be performed in my opinion on animal models in an in vivo setting where physiological motion is also present.

We appreciate that the original submission was lacking in a clear demonstration of the true potential of HySE in a real-time endoscopic imaging scenario. Therefore, we performed hyperspectral imaging in an animal model as suggested by the reviewer (Fig. 8; Supplementary Movie 3). In order to mimic a realistic clinical imaging condition, an intact ex vivo porcine oesophagus and clinical gastroscope were exploited. The HySE with internal illumination method was inserted into the accessory channel of the gastroscope, and the imaging area of the HySE was controlled by bending the gastroscope laterally. Although there was no physiological motion in the ex vivo porcine oesophagus, random motions were produced due to movements of the clinical gastroscope mimicking a clinical imaging condition with random motion.

We measured hyperspectral imaging within the ex vivo porcine oesophagus with an imaging speed over 20 fps using the internal illumination method before and after methylene blue (MB) staining. The successful reconstruction of the hypercube presenting images with clear morphological delineation of the lumen suggests that applied corrections are accurate. Furthermore, the spectral information of the lumen agreed with that expected, particularly after application of the MB staining. Moreover, the concentration of MB staining in the lumen could be quantified via linear spectral unmixing. This experiment is now presented in the Results (p12) and Methods (p23), as well as Supplementary Movie 3.

Table 1 in the revised manuscript shows the summary of specifications of the HySE, which clearly has significant technical improvements against previous hyperspectral imaging endoscopy. Furthermore, direct demonstration of the system operation in the presence of gastroscope motion in a pig oesophagus lumen provides compelling evidence of the future system potential. We therefore believe that our optimised HySE system demonstrates sufficient merit for publication in Nature Communications

Reviewer #3 (Remarks to the Author):

The authors described a real-time hyperspectral endoscope they have developed that is claimed to overcome the present challenges of slow image acquisition and distortions during free-hand operation. Such an instrument would bring a new capability to medical screening and diagnostics since it could capture chemical signatures embedded in the reflection spectrum, or fluorescence spectrum, that are biomarkers of disease.

We would like to thank the reviewer for their careful reading of the manuscript. We hope that the changes that we have made in our revised manuscript are sufficient to satisfy their concerns, in particular with regard to the concerns regarding: the spectral resolution and bandwidth; the limitations of internal illumination; the demonstration of freehand operation in a realistic scenario; and lack of details in some methods.

p3 line 81. please change "imaging mapping" to "image mapping"

This has been corrected.

p3 line 95 please change to "CE-marked"?

This has been corrected.

p13, line 326. For these simulations, just three wide bandwidth spectral images were employed. It is unclear if this provides enough wavelengths and enough spectral resolution to give a prediction for later development as a full hsi endoscope. Please provide planned path for testing at high spectral resolution and describe how the high resolution wavelength axis used in later simulations was created. What kind of calibration source was used. Linear or poly model fitted to recorded spectrum?

We apologise for the previous lack of details regarding our estimation of spectral resolution and methods for creating a wide range spectral image. We updated the manuscript to address these omissions.

For spectral resolution, Mercury (250 – 550 nm) and Neon-Argon (585 – 965 nm) lamps that emit multiple peaks with very narrow bandwidth (less than 0.2 nm) at specific wavelengths in a calibration light source were exploited. Because the bandwidth of each peak is very narrow, we defined a full-width at half maximum of measured signals as spectral resolution (per reference 2 in the revised manuscript). These details have been added to the Results (p8) and Methods (p20).

For the spectral bandwidth, this is dependent on the number of lines in the grating. Thus depending on the grating, multiple spectral images may have to be measured at different grating angles to create a wide range spectral image. These details have been added to the Methods (p16) and comparison of data acquired using 3 different gratings have been added to the Results (p8) and Supplementary Figure 3.

p13, line 337. How will limited internal illumination be overcome in a prototype, where the target is out of range of external illumination?

With further optimisation of the optical setup for the internal illumination method we have been able to achieve optical power at the sample plane of 12.00 ± 0.43 mW. Using this illumination, we have included an additional experiment in the manuscript demonstrating that HySE enables hyperspectral imaging of an intact porcine oesophagus with imaging speed over 20 fps using the internal illumination method. This indicates that internal illumination will not be a limitation in clinical applications within a lumen.

p14, line 360. How are radial distances known during rigid or freehand imaging, and can the tolerance for error be described? Was it possible to get a correspondence between radial distance and the magnification in the geometric transformation matrix?

The correction of lens distortion is performed using a correcting coefficient α , which is assumed to be constant over the narrow range of working distances used. The purpose of the pre-processing is to correct the wide-field images to achieve the highest accuracy in wide-field image registration. Thus, we can use radial distances of specific pixels from the image centre for the correction of lens distortion rather than actual radial distances. The geometric transformation matrix only provides relative information about image distortion among wide-field images, so we do not consider exact correspondence between radial distance and magnification.

p17, line 420. This describes a freehand imaging evaluation using a planar phantom drawn on paper. The critical problem is to be able to collect and render an hsi image from intravital space, usually convoluted and surrounding the endoscope. Thus it is not clear how well the authors' evaluation can predict performance under actual circumstances. Tissue phantoms and samples described later give a better prediction of the method of hsi endoscopy, yet this appears to still be an external imaging setup with the external illuminator. This reviewer believes additional tests of the endoscope using an animal are needed in order to justify a move to a clinical setting. Suggest working in the rat, using the GI surface or folds in the mouth area. Do the tissue reflectance spectra obtained compare well to known spectra, and are the geometric and light shading corrections adequate?

This is an excellent point made by the reviewer. As the reviewer suggested, we have now performed hyperspectral endoscopy in an animal model. In order to mimic a realistic clinical imaging condition, an intact ex vivo porcine oesophagus and clinical gastroscope were exploited. The HySE with internal illumination method was inserted into the accessory channel of the gastroscope, and the imaging area of the HySE was controlled by bending the gastroscope laterally. Although there was no physiological motion in the ex vivo porcine oesophagus, random motions were produced due to movements of the clinical gastroscope mimicking a clinical imaging condition with random motion.

We measured hyperspectral imaging within the ex vivo porcine oesophagus with an imaging speed over 20 fps using the internal illumination method before and after methylene blue (MB) staining. The successful reconstruction of the hypercube presenting images with clear morphological delineation of the lumen suggests that applied corrections are accurate. Furthermore, the spectral information of the lumen agreed with that expected, particularly after application of the MB staining. Moreover, the concentration of MB staining in the lumen could be quantified via linear spectral unmixing. This experiment is now presented in the Results (p12) and Methods (p23), as well as Supplementary Movie 3.

Fig 1. The caption describes an internal fiber illuminator however also calls out either external or internal illumination. The external method is illustrated in the figure. Most of the simulation used external fiber illumination, however the two cases would very likely need different light/dark calibration. Please make sure the illumination method is given for each of the system tests.

White and dark reference images are measured in every experiment because they vary according to experimental conditions. We added this information in the Methods (p18).

Fig. 2. In B, only translation appears, would it be better to include rotation and magnification examples in the part? In all, the hsi image generation appears to be similar to pushbroom scanning if translation is the only variation. It isn't clear here how rotation and magnification are built into the geo-correction process. Apparently it would require the 2-D reference image.

As reviewer suggested, we revised Fig 2b and added Supplementary Movie 1 to clarify the wide-field image registration process, along with a flow chart in Supplementary Figure 1.

Fig. 3. In A, it shows one acquired image and three variants made from the first, is this correct?.

The four images illustrate different types of image deformation used in simulation. We have added Supplementary Movie 1 to help readers in understanding how these images are prepared.

B is presumably a co-registered image where the rectangle covers the shared, or common, areas from all of the input images. C, missing letters result from spectral differences, but this should be reiterated in the text. Unclear how a.u units are established in the scale bars.

The reviewer is correct. In C, features are absent from the *in silico* USAF target image when there is no spectral absorbance defined at the simulated wavelength. This information has been added to the Results (p7). In the simulation, the absorption spectra of the 6 spectral features have values between 0 and 1 to emulate the experimental condition. The absorption and reflectance values in hypercube were obtained by dividing measured images by white reference images obtained in the same system, thus the result is dimensionless and we have removed the arbitrary unit designation in the scale bars.

D. Not clear what is different about the area enclosed by rectangle, more explanation of this process is needed.

We appreciate the reviewer pointing out this oversight. The white-dashed squares indicate the slit scanning regions where hyperspectral imaging is performed. We have added this description in the revised figure legend.

E. the images red and blue appear to be identical in the figure. Is that correct?

We are not exactly clear on the red and blue that the reviewer is referring to. Nonetheless, we assume that this question relates to the different intensity levels of the images; the USAF target reflects most of light signal so the normalised images using a white background image seem to be identical. However, as demonstrated in the figure below, pre-processing for intensity normalisation is required.

Figure for the reviewer. Representative slices of hypercube before (top) and after (bottom) normalisation process at four different wavelength.

Fig. 6.

A. Could not see the white dashed line in the submission.

This has been removed and the false colour area is now denoted in the Figure legend.

B. These images look promising.

Thank you for this positive comment.

Please give method for a.u. determinations in B and C. D.

The absorption and reflectance values were obtained by dividing measured signals by reference signals thus it is always dimensionless. The arbitrary unit designations have therefore been removed. We thank the reviewer for highlighting this issue.

Change "blood filled" to "blood-filled".

This has been corrected.

Fig S2. B. Have wavelength values been associated with the simulation band numbers in x axis? If so please provide and describe the calibration.

The x-axis in Fig S2. B indicates spectral channel numbers in simulation. Each spectral feature has different absorption values over spectral channels (x-axis) and no calibration was performed against wavelength because 6 spectral features were randomly generated

Fig S6. Method for generating reflectance scale bars was not stated. How is the value of the reflectance determined?

The reflectance values were determined not to make the saturated synthetic RGB image. We added this information in the Methods (p19).

Other. The instrument description seems to show that an OEM or custom built spectrograph system was used. Authors mention checking the spectral resolution with a calibration data from a reference source. Please specify the spectral properties of the calibration source, i.e., were Xe or Ar gas calibration lamps used which provide narrow spectral lines at different wavelengths? The method for determining the hsi wavelength axis should also be provided, even if a reference to these details was given. In certain places there are scale bars noting optical quantities such as a.u. etc. Please give more detail how these were arrived at.

As reviewer suggested, we added details of light sources for estimating spectral resolution and other associated information to the Results (p8) and Methods (p20).

REVIEWERS' COMMENTS:

Reviewer #1 (Remarks to the Author):

Firstly, I would like to commend the authors for the significant improvements in the manuscript, including new experiments and demos, new movies, and the thorough incorporation of reviewer comments and questions.

A minor point (maybe): Figure 8b and 8e appear to be mirrored – are they supposed to be in the same orientation?

I now suggest this article be published.

Charles H. Camp Jr.

Reviewer #2 (Remarks to the Author):

I think the authors have addressed the comments given by the reviewers and mine, and have improved the content over the previous version of the manuscript.

Now having said that, I still think this work represents an incremental advancement over other similar works. The main message of this work (as I understand it from the original submission) is to use “white light” acquired data to compensate for the motion occurring during “spectral data” acquisition in order to provide artifact free images. While interesting, in my opinion this does not represent a sufficiently innovative and/or complex approach over concurrent similar setups. As such I think it will be more indicated for a journal where similar works have been published so far and where innovations of similar entity and nature have been introduced in the past.

Reviewer #3 (Remarks to the Author):

The manuscript is much improved. In the authors' rebuttal their definition for reflectance is correct but of course there is a different equation for absorption. Authors are urged to provide these in text. For some reason the authors did not describe the procedure for their spectrograph wavelength calibration, presumably carried out with a gas calibration lamp. Their characterizations of spectral resolution are clear and well written. But it is important to show the procedure for obtaining an accurate wavelength axis. Accuracy to the level of one decimal place is reported in results. Presumably the calibration was done by a curve fitting procedure with a gas calibration lamp. Please elaborate or correct. The only other criticism is that the authors appear not to have provided a comparison of reflectance spectra obtained from their ex vivo animal model with a known reference for the same tissue model. This is needed to confirm that the authors results are comparable with known literature. The SAM analysis between disease classes was well written and the rest of the revision is much improved.

REVIEWERS' COMMENTS:

Reviewer #1 (Remarks to the Author):

Firstly, I would like to commend the authors for the significant improvements in the manuscript, including new experiments and demos, new movies, and the thorough incorporation of reviewer comments and questions.

A minor point (maybe): Figure 8b and 8e appear to be mirrored – are they supposed to be in the same orientation?

I now suggest this article be published.

Charles H. Camp Jr.

We would like to thank the reviewer for their time in carefully reading the manuscript. According to the minor point about figures 8b and 8e, we measured the pig oesophagus before (fig 8b) and after (fig 8e) methylene blue staining. The scanning angle and direction of HySE in the pig oesophagus were determined according to the orientation of the clinical gastroscope. Thus while the images appear to be mirrored, they were actually simply measured in different scanning directions. We added this information in the Methods (p23).

Reviewer #2 (Remarks to the Author):

I think the authors have addressed the comments given by the reviewers and mine, and have improved the content over the previous version of the manuscript.

Now having said that, I still think this work represents an incremental advancement over other similar works. The main message of this work (as I understand it from the original submission) is to use “white light” acquired data to compensate for the motion occurring during “spectral data” acquisition in order to provide artifact free images. While interesting, in my opinion this does not represent a sufficiently innovative and/or complex approach over concurrent similar setups. As such I think it will be more indicated for a journal where similar works have been published so far and where innovations of similar entity and nature have been introduced in the past.

We appreciate the constructive comments of the reviewer.

Reviewer #3 (Remarks to the Author):

The manuscript is much improved. In the authors' rebuttal their definition for reflectance is correct but of course there is a different equation for absorption. Authors are urged to provide these in text.

We would like to thank the reviewer for their careful reading of the manuscript. As the reviewer suggested, we have revised the Methods (p19) to clarify the equations used to calculate reflectance and absorbance from the hypercube and added appropriate references.

For some reason the authors did not describe the procedure for their spectrograph wavelength calibration, presumably carried out with a gas calibration lamp. Their characterizations of spectral resolution are clear and well written. But it is important to show the procedure for obtaining an accurate wavelength axis. Accuracy to the level of one decimal place is reported in results. Presumably the calibration was done by a curve fitting procedure with a gas calibration lamp. Please elaborate or correct.

As reviewer suggested, we added details of the curve fitting for estimating spectral resolution to the Methods (p17). The curve fitting was performed by the commercial software (Lightfield v6.7, Princeton Instrument) based on the Czerny-Turner model and the full width at half maximum of the curve fitting result is given as the spectral resolution of HySE.

The only other criticism is that the authors appear not to have provided a comparison of reflectance spectra obtained from their ex vivo animal model with a known reference for the same tissue model. This is needed to confirm that the authors results are comparable with known literature.

There are little known references that measure hyperspectral signals in the pig oesophagus. To validate the measured absorbance via HySE, we measured ground-truth absorbance of the normal and methylene-blue stained pig oesophagus tissue using a spectrometer. These data are now included as Supplementary Fig 8 c-e. The measured absorbance via HySE was consistent with the ground-truth absorbance. We added details of this validation information to the Results (p12).

The SAM analysis between disease classes was well written and the rest of the revision is much improved.

Thank you for this positive comment.